# FedSFT: Resource-Constrained Federated Black-Box Adaptation of Large Language Models

## Abstract

Federated fine-tuning enables privacy-preserving adaptation of large language models (LLMs) by allowing decentralized training without sharing raw data. However, its real-world deployment is often hindered by restricted access to model parameters and substantial computation, communication, and memory overhead. To address these challenges, we propose **Fed**erated **S**urrogate **F**ine-**T**uning (FedSFT), a novel framework for federated black-box fine-tuning of LLMs that requires access only to the token probabilities of output sequences and significantly reduces resource demands on clients. In each communication round of FedSFT, clients fine-tune a small model that serves as a surrogate for the large model hosted on the server. The server then leverages the logit offsets between the tuned and untuned small models to adjust the output of the untuned large model and distills the knowledge to update the small model for the next training round. Experimental results show that FedSFT significantly reduces client-side computation, communication, and memory overhead while maintaining competitive performance compared to direct federated fine-tuning of large models. FedSFT offers a promising solution for efficient and privacy-preserving black-box fine-tuning of large models on resource-constrained clients, broadening the accessibility and applicability of state-of-the-art LLMs.

## 1 Introduction

Recent advancements in large language models (LLMs), exemplified by models like LLaMA 3 (Touvron et al., 2023) and GPT-4 (Achiam et al., 2023) trained on massive, diverse public datasets with up to hundreds of billions of parameters, have demonstrated remarkable zero-shot and few-shot learning across various language tasks, including text generation, question answering, and machine translation. Fine-tuning these untuned, general-purpose LLMs on task-specific datasets is a common approach to achieve desired performance (e.g., PMC-LLaMA (Wu et al., 2023) for improved accuracy on medical questions). However, the practical application of LLM fine-tuning often faces the challenge of task-specific data (such as patient medical records) being distributed across multiple locations, making centralization expensive and potentially compromising patient privacy.

To overcome this issue, federated learning (FL) (McMahan et al., 2017) that enables collaborative model training without sharing raw data is the state-of-the-art solution. However, directly fine-tuning LLMs in FL faces several major challenges: 1) **Computation and communication overhead:** The federated fine-tuning process for LLMs involves significant computational and communication costs due to the extensive number of trainable parameters in LLMs. Clients with limited computational power may find it challenging to carry out complex model updates, resulting in prolonged training time. Transferring a large number of model parameters between the server and clients also incurs high communication costs, consuming considerable bandwidth and increasing latency. 2) **Memory footprint:** Local fine-tuning of LLMs on each client requires substantial on-device memory to store not only trainable model parameters, but also their gradients and optimizer states that can exceed the size of the parameters themselves. For instance, fine-tuning a LLaMA 7B model directly with a single batch size demands at least 58 GB of memory (14GB for trainable parameters, 42GB for Adam optimizer states and weight gradients, and 2GB for activations) (Zhao et al., 2024), making it impractical for resource-constrained clients to participate in federated fine-tuning of LLMs.

3) **Access to full model parameters:** The most powerful versions of LLMs, such as GPT-4.5 and Gemini 2.5, are proprietary, with their model parameters unavailable to the public. Consequently, clients cannot perform local fine-tuning of these models within FL. Thus, novel approaches that do not require full model access are both desirable and necessary.

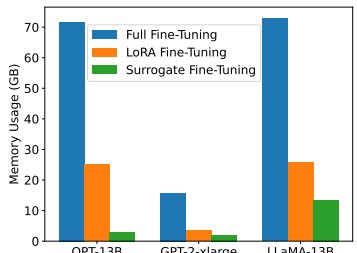

To overcome these challenges, parameter-efficient fine-tuning (PEFT) methods, including prompt tuning (Lester et al., 2021), adapter tuning (Houlsby et al., 2019), and low-rank adaptation (LoRA) (Hu et al., 2021), have been integrated into FL. These methods freeze most parameters of untuned LLMs and update only a few trainable parameters, leading to a significant reduction in computation and communication overhead. However, since the gradient computation for trainable parameters needs back-propagation through the entire LM, fine-tuning with PEFT still requires a significant memory footprint, as shown in Figure 1. For instance, fine-tuning a LLaMA 13B model with LoRA and a batch size of 1 demands around 25.8 GB of memory, exceeding the capacity of most resource-constrained edge devices ($4 \sim 8$ GB), such as smartphones and IoT devices. Furthermore, these methods still require the clients to access full model parameters and gradients for local training.

Figure 1: Peak memory footprint of different training methods for each client with batch size of 1 and sequence length 512.

To address the challenges of fine-tuning black-box LLMs in federated settings, we propose **Fed**erated **S**urrogate **F**ine-**T**uning (FedSFT), a novel framework that significantly reduces the computation, communication, and memory overheads on resource-constrained clients. FedSFT is inspired by the observation that behavioral changes of language models during fine-tuning reflect critical model capabilities (Liu et al., 2021; Mitchell et al., 2023). Therefore, instead of directly fine-tuning the target large model, clients in FedSFT collaboratively fine-tune a small surrogate model. The knowledge from this tuned small model is then integrated with the untuned large model to emulate the behavior of a fully tuned large model. Figure 2 depicts the overall FedSFT process. Initially, the server hosts a pair of untuned large and small models that share the same tokenizer and distributes the small model to each client. In each training round, clients locally fine-tune their small models on private data using LoRA, and send the LoRA updates back to the server. The server aggregates the received LoRA updates to obtain a globally tuned small model and constructs a composite model consisting of three components: the tuned LoRA modules, the untuned small model, and the untuned large model. This composite model is then used for knowledge distillation to update the small model for the next training round. Specifically, the server computes logit offsets, the differences between the tuned and untuned small models, and uses them to adjust the untuned large model's outputs, and then further tunes the small model to align with these adjusted outputs using a small public dataset.

Evaluation across multiple datasets and LLMs demonstrates that FedSFT achieves performance comparable to direct federated fine-tuning method, despite lacking access to full model parameters. Specifically, with a batch size of 1 and an input length of 512 tokens, FedSFT reduces communication overhead, computation cost, and memory usage by over $4.2\times$, $9.6\times$, and $8.3\times$, respectively, by fine-tuning OPT-1.3B instead of OPT-13B. This enables resource-constrained clients to participate in federated fine-tuning, making FedSFT a promising step toward democratizing LLM.

In summary, this paper makes the following main contributions:

- We propose FedSFT, a novel framework that enables resource-constrained clients to collaboratively fine-tune a black-box LLM without accessing the model parameters or sharing the private data.

- We develop a novel strategy in FedSFT that can effectively integrate the knowledge from the untuned large model and tuned small model to guide the local fine-tuning of clients in each FL training round.

- We conduct extensive experiments across diverse models, tasks, and datasets, demonstrating that FedSFT achieves comparable model performance to direct federated fine-tuning of LLMs, while significantly reducing computation, communication, and memory overheads.

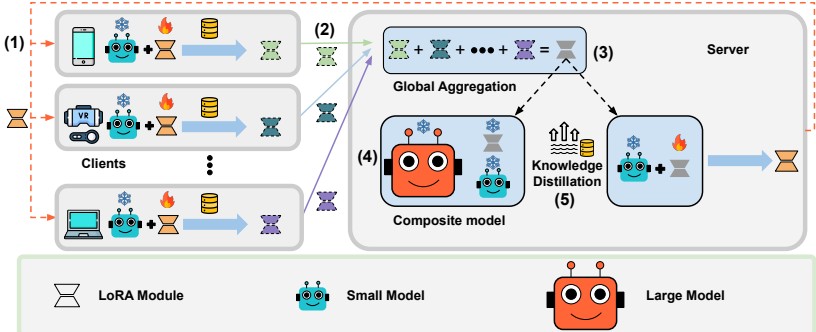

Figure 2: Overview of FedSFT. Each training round comprises the following steps: 1) The server broadcasts the latest LoRA weight matrices of the small model to all clients; 2) Each client fine-tunes the small model's LoRA weight matrices and sends the updated LoRA weight matrices to the server; 3) The server aggregates the updated local LoRA weight matrices to obtain the global LoRA weight matrices; 4) The server constructs the composite model; and 5) The server updates the global LoRA weight matrices to align its predictions with the composite model via knowledge distillation.

## 2 RELATED WORK

Existing literature on federated fine-tuning of LLMs can be broadly categorized based on the adopted PEFT methods. For example, federated prompt tuning has been studied in Zhao et al. (2023b;a); Che et al. (2023), where clients update only a small set of soft prompt parameters while keeping the base LLM frozen. However, these approaches still require backpropagation through the entire LLM to update the prompts, leading to high resource consumption and necessitating white-box access to the model. In contrast, recent methods such as Fed-BBPT (Lin et al., 2023) and FedBPT (Sun et al., 2023) support prompt tuning in black-box settings, where clients interact with the LLM solely via APIs without access to model internals. Nonetheless, prompt tuning modifies only the model's input representations without updating its internal weights, which may limit effectiveness on complex tasks that require substantial weight adaptation. As one of the most popular PEFT methods, LoRA has been incorporated into various federated fine-tuning schemes for LLMs (Zhang et al., 2024; Sun et al., 2024; Bai et al., 2024; Wang et al., 2024). Nonetheless, these approaches face similar challenges as federated prompt tuning, including high memory overhead on clients and the need for full access to model parameters. To mitigate memory usage, some works (Ling et al., 2024; Xu et al., 2024; Qin et al., 2024) introduce zeroth-order optimization during local fine-tuning, which avoids backpropagation. However, these methods still require full model deployment on the client side, making them inapplicable to the black-box setting considered in this paper.

A few recent studies (Wu et al., 2024; Peng et al., 2024) have applied the offsite-tuning (OT) (Xiao et al., 2023) to FL by constructing a small submodel for local fine-tuning at clients, which is then used to guide the tuning of the target LLM at the server. While sharing some conceptual similarities, our approach differs fundamentally from OT in two key aspects. First, OT constructs a small sub-model by truncating layers from the target LLM, and then performs model compression on a large corpus to align its outputs with those of the target LLM. This compression step incurs additional computational and data costs. In contrast, our approach allows the small model to be any untuned language model as long as it shares the same tokenizer as the target LLM, eliminating the need for costly model compression. Second, OT limits the compression ratio to relatively modest values (e.g., 0.5 in Wu et al. (2024)) to maintain performance, whereas our approach supports small models that are up to 10× smaller than the target LLM, offering significantly greater scalability.

## 3 FEDSFT: FEDERATED SURROGATE FINE-TUNING

### 3.1 PROBLEM SETTINGS

Consider an FL system with a server and a set of $K$ clients. Each client $k \in [K]$ has a local private dataset $\mathcal{D}_k$ for downstream tasks such as natural language understanding or generation. The server hosts a black-box untuned LLM $\mathcal{M}_l$ that provides output logits over the entire vocabulary for any given input, without revealing its internal model parameters. Our goal is to tune the untuned LLM

---

**Algorithm 1** Proposed FedSFT Algorithm

---

**Input:** Untuned large model $\mathcal{M}_l$ and small model $\mathcal{M}_s$, total training rounds $T$, public dataset $\mathcal{D}_{kd}$
**Output:** Composite model $\tilde{\mathcal{M}}_l$

---

1: Initialize LoRA weight matrix $A$ with random Gaussian values and matrix $B$ with zeros
2: **for** round $t = 0, 1, 2 \ldots, T - 1$ **do**
3:    Server sends $(A, B)$ to all clients
4:    **for** each client $k \in [K]$ **in parallel do**
5:       $(A_k, B_k) \leftarrow$ Update $(A, B)$ by solving (3)
6:       Send $(A_k, B_k)$ back to the server
7:    **end for**
8:    Server aggregates the LoRA weight matrices according to (4)
9:    Server obtains the surrogate-tuned outputs for the data samples in $\mathcal{D}_{kd}$ according to (7)
10:    Server updates $(A, B)$ by knowledge distillation according to (8)
11: **end for**
12: **return**  $\tilde{\mathcal{M}}_l := (\mathcal{M}_l, \mathcal{M}_s, A, B)$

---

$\mathcal{M}_l$ with model parameters $W_l$ on the union of all local datasets $\mathcal{D} := \bigcup_{k \in [K]} \mathcal{D}_k$ by solving the following optimization problem:

$$\min_{\Delta W} f(\Delta W) := \sum_{k=1}^{K} p_k f_k(\mathcal{M}_l(W_l + \Delta W)), \tag{1}$$

where $f_k(\mathcal{M}_l(W_l + \Delta W)) = \mathbb{E}_{z \in \mathcal{D}_k}[\ell(\mathcal{M}_l(W_l + \Delta W); z)]$ is the local objective function of client $k$ with $\ell(\cdot; z)$ being the model loss on a data sample $z$, and $p_k$ denotes the weight coefficient for client $k$. In our setting, the clients are unwilling to share their privacy-sensitive data with the server and are also resource-constrained, lacking the capacity to train the target LLM locally due to the high memory demands and significant communication and computation costs of fine-tuning.

In this paper, we propose a novel approach called FedSFT, where each client only fine-tunes a small model to steer $\mathcal{M}_l$ to act like a tuned model on all client datasets while avoiding the cost of tuning $\mathcal{M}_l$ directly. As depicted in Figure 2, FedSFT consists of two main components: *client-side LoRA fine-tuning of small models*, and *server-side black-box large model adaptation*. In the following, we will describe the design strategies for these two components in detail. The pseudocode of FedSFT is provided in Algorithm 1. It is worth noting that FedSFT preserves the privacy benefits of standard FL by keeping raw data local and sharing only model parameters. It is also compatible with existing privacy-preserving FL techniques such as secure aggregation and differential privacy.

## 3.2 CLIENT-SIDE LORA FINE-TUNING OF SMALL MODELS

To address the substantial resource overheads and white-box access requirements associated with directly fine-tuning the large model $\mathcal{M}_l$, clients in FedSFT collaboratively fine-tune an untuned small model $\mathcal{M}_s$ using LoRA. The small model $\mathcal{M}_s$ shares the same tokenizer and vocabulary as the large model $\mathcal{M}_l$. The resulting tuned model, $\mathcal{M}_s^+$, is then used to steer the outputs of $\mathcal{M}_l$ to closely approximate those of its directly tuned counterpart. Specifically, let $W_s \in \mathbb{R}^{m \times n}$ denote the weight matrix of small model $\mathcal{M}_s$ for the target module (e.g., the query, key, value projection layers in self-attention layers), and $B \in \mathbb{R}^{m \times r}$, $A \in \mathbb{R}^{r \times n}$ denote the corresponding trainable low-rank matrices injected into this module. Then the weight matrix of the tuned small model $\mathcal{M}_s^+$ becomes

$$\hat{W}_s = W_s + BA. \tag{2}$$

Here, the rank $r \ll \min\{m, n\}$, thereby reducing the number of trainable parameters compared to full fine-tuning.

During each FL round, after receiving the latest global LoRA weight matrices $B, A$ of the small model from the server, each client $k \in [K]$ freezes the base model weight matrix $W_s$ and updates the LoRA weight matrices as follows:

$$(A_k, B_k) = \arg\min_{A,B} \sum_{z \in \mathcal{D}_k} f_k(\mathcal{M}_s(W_s + BA); z). \tag{3}$$

By applying LoRA for local fine-tuning of the lightweight small model, we significantly reduce computation and memory overheads, since only a small number of LoRA parameters are updated while keeping the base model frozen. Furthermore, each client uploads only the LoRA matrices $(A_k, B_k)$ to the server rather than the entire model, thereby reducing communication costs.

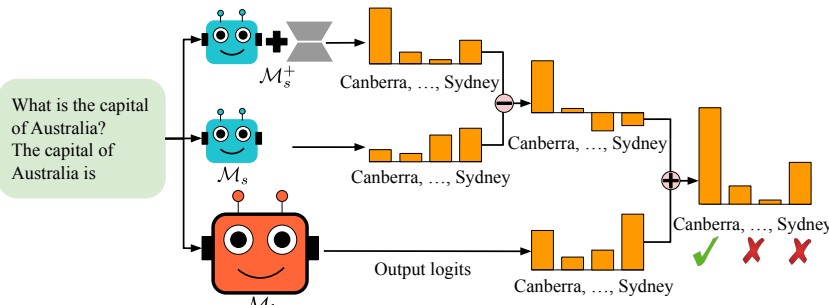

Figure 3: Inference with composite model $\tilde{\mathcal{M}}_l$.

## 3.3 SERVER-SIDE BLACK-BOX LARGE MODEL ADAPTATION

Upon receiving the LoRA weight matrices $\{(A_k, B_k)\}_{k \in [K]}$ from all clients, the server aggregates them to obtain the global LoRA weight matrices $(A, B)$ as follows:

$$A \leftarrow \sum_{k \in [K]} p_k A_k, \qquad B \leftarrow \sum_{k \in [K]} p_k B_k. \tag{4}$$

Before sending the aggregated LoRA weight matrices $(A, B)$ back to the clients for further training as in the traditional federated LoRA fine-tuning methods, we draw inspirations from prior works (Li et al., 2022; Liu et al., 2024; Mitchell et al., 2023) on the information gained from pre-training and fine-tuning in LLM: pre-training at scale enables greater accumulation of raw knowledge (improved factual correctness), while fine-tuning at larger scale produces greater helpfulness (improved user satisfaction) (Gudibande et al., 2023). Therefore, we could boost the performance of the small tuned model $\mathcal{M}_s^+$ by ensembling it with the large untuned model $\mathcal{M}_l$, which will better guide the local training of all clients in the next round.

To achieve that goal, *the server constructs a composite model $\tilde{\mathcal{M}}_l$ that mimics the performance of a directly tuned version of $\mathcal{M}_l$*. We formulate fine-tuning $\mathcal{M}_l$ as a reinforcement learning (RL) problem, where the objective is to maximize task reward while incorporating a KL-divergence penalty to constrain deviation from a reference (large untuned) model. Specifically, following Rafailov et al. (2023); Jaques et al. (2020), the RL objective can be written as

$$\tilde{\mathcal{M}}_l = \arg\max_{\mathcal{M}_l^+} \mathbb{E}_{x \sim \mathcal{D}, y \sim \mathcal{M}_l^+(\cdot|x)}[r(x, y) - \beta \text{KL}(\mathcal{M}_l^+(y|x) || \mathcal{M}_l(y|x))], \tag{5}$$

where $r(x, y)$ represents the reward, $\mathcal{M}_l^+$ denotes the large tuned model, $\mathcal{M}_l$ is the large untuned model, and $\beta$ is a parameter that controls the strength of the KL constraint toward $\mathcal{M}_l$. The optimal solution to (5) is given by

$$\tilde{\mathcal{M}}_l(y|x) = \frac{1}{Z(x)} \mathcal{M}_l(y|x) \exp\left(\frac{1}{\beta} r(x, y)\right), \tag{6}$$

where $Z(x) = \sum_y \mathcal{M}_l(y|x) \exp\left(\frac{1}{\beta} r(x, y)\right)$ is the partition function (see detailed derivation in Appendix A.1). Since the large model is a black-box and cannot be directly fine-tuned, we instead fine-tune a small model on the downstream task. To capture the change in output probabilities induced by the downstream dataset, we define the reward as $r(x, y) = \beta \log \frac{\mathcal{M}_s^+(y|x)}{\mathcal{M}_s(y|x)}$, which reflects the distributional shift between $\mathcal{M}_s^+$ and $\mathcal{M}_s$. This reward is then integrated with the large untuned model to construct the surrogate fine-tuned model $\tilde{\mathcal{M}}_l$, as formalized in the following theorem.

**Theorem 1.** *Given the RL objective in (5), when setting reward as $r(x, y) = \beta \log \frac{\mathcal{M}_s^+(y|x)}{\mathcal{M}_s(y|x)}$, the probability distribution of the next token prediction from the composite model $\tilde{\mathcal{M}}_l$ is given by:*

$$\pi_{\tilde{\mathcal{M}}_l}(X_t|x_{<t}) := softmax\left[g_{\mathcal{M}_l}(X_t|x_{<t}) + \alpha\left(g_{\mathcal{M}_s^+}(X_t|x_{<t}) - g_{\mathcal{M}_s}(X_t|x_{<t})\right)\right], \tag{7}$$

*where $x_{<t} := \{x_0, \ldots, x_{t-1}\}$ is given the input token sequence at each time step $t$. $g_{\mathcal{M}_l}$, $g_{\mathcal{M}_s^+}$, and $g_{\mathcal{M}_s}$ are the logit scores over the vocabulary produced by the language modeling heads of models $\mathcal{M}_l$, $\mathcal{M}_s^+$, and $\mathcal{M}_s$, respectively. The hyperparameter $\alpha$ controls the influence on the output distribution of the $\mathcal{M}_l$.*

*Proof.* See Appendix A.2. □

As illustrated in Figure 3, the composite model generates the next token by combining the outputs from the three models $\mathcal{M}_l$, $\mathcal{M}_s$, and $\mathcal{M}_s^+$. We then use knowledge distillation to transfer the knowledge from the composite model $\tilde{\mathcal{M}}_l$ (i.e., as the teacher) to the small tuned model $\mathcal{M}_s^+$ (i.e., as the student). For the distillation, the teacher model is evaluated on samples of *unlabeled data* on the server based on (7), and their logit outputs are used to further tune the LoRA weight matrices of the student model on the server:

$$(A, B) = \arg\min_{A,B} \mathbb{E}_{x \sim \mathcal{D}_{kd}} \left[ \mathrm{KL} \left( \pi_{\tilde{\mathcal{M}}_l}(x) || \mathrm{softmax}\big(g_{\mathcal{M}_s^+}(x)\big) \right) \right], \tag{8}$$

where $\mathcal{D}_{kd}$ is a small public unlabeled dataset that could come from other domains as demonstrated in our experiments, and $g_{\mathcal{M}_s^+}(x) := g(\mathcal{M}_s(W_s + BA); x)$ is the logits output of the small tuned model given input $x$.

After obtaining the newly updated global LoRA weight matrices from (8), the server distributes them to the clients for the next round of training. During the inference, the client sends a query to the server, which encodes it into a prompt token sequence and generates the response sequence according to (7). This requires computing one forward pass of the large model and two forward passes of the small model (one for the untuned small model, and the other for the tuned small model). When the size of the small model is much smaller than that of the large model, the total inference cost is comparable to that of using the directly tuned large model.

## 4 EXPERIMENTS

In this section, we detail our experimental setup, including the tasks, datasets, models, baseline methods, and evaluation protocol. We then present the main results of FedSFT on each task, followed by an analysis of system costs. Finally, we conduct ablation studies to assess the impact of key design components in FedSFT. Additional results and analyses are provided in the Appendices.

### 4.1 EXPERIMENTAL SETUP

We empirically evaluate our method in a cross-silo federated setup involving 10 clients, focusing on two tasks: controlled sentiment generation and a more challenging instruction-following task. All experiments are conducted on a server with 4 A6000 GPUs. Hyperparameters are detailed in Appendix B.5. Our code and data are publicly available at `https://anonymous.4open. science/r/FedSFT-F808`.

**Tasks and Datasets.** For the controlled sentiment generation task, we use the "imdb-preference" dataset[1], which consists of 20,000 training data points, each containing a prompt and a pair of responses scored by a golden reward model (Zhou et al., 2024). We allocate 128 examples for knowledge distillation and split the remaining data across 10 clients for training. For evaluation, we randomly select 200 samples from the test set. For the instruction-following task, we adopt the "databricks-dolly-15K" dataset (Ouyang et al., 2022), which includes over 15,000 instruction-response examples. For knowledge distillation in this task, we use subsets of the Alpaca dataset (Taori et al., 2023) (128 examples for GPT-2 and 512 for OPT/LLaMA), selected for their diversity and accessibility. To simulate an FL setup similar to FedIT (Zhang et al., 2024), we adopt two data partition strategies: pathological non-IID (McMahan et al., 2017) and Dirichlet non-IID (Hsu et al., 2019). Due to the page limit, we present the results on pathological non-IID distribution in the main paper, and the results on the Dirichlet distribution in Appendix B.2. We evaluate on three distinct test sets: 1) **Dolly**: 500 samples from databricks-dolly-15K. 2) **SelfInst** (Wang et al., 2023): 252 user-generated instructions. 3) Super-NaturalInstructions (**S-NI**)(Wang et al., 2022): 9,000 samples across 119 tasks. Following (Peng et al., 2023; Gu et al., 2023), we divide this set into three subsets based on ground truth response lengths: [0, 5], [6, 10], and [11, $+\infty$], and use the [11, $+\infty$] response length subset for evaluation. Details on datasets are in Appendix B.1.

**Models.** For each task, we employ pairs of small models and large models. For controlled-sentiment generation, we employ the GPT-2-base (120M)[2] as the small model and GPT-2-xlarge

---

[1] `https://huggingface.co/datasets/ZHZisZZ/imdb_preference`
[2] `https://huggingface.co/lvwerra/gpt2-imdb`

(1.5B) as the large model. For instruction-following, we use the following pairings of small model and large model: OPT-1.3B with OPT-13B, GPT-2-large (760M) with GPT-2-xlarge (1.5B), and LLaMA-7B with LLaMA-13B.

**Baselines.** We compare FedSFT with the following baselines: 1) **Base** directly uses the large model for zero-shot inference. 2) **FedIT** (Zhang et al., 2024) fine-tunes the small model or large model with LoRA by the FedAvg algorithm (McMahan et al., 2017). 3) **FedIT+SFT** follows the same procedure as FedIT to fine-tune the small model. During inference, it utilizes a composite model consisting of the tuned and untuned small models and the large model to generate responses. 4) **FedOT**, the federated version of Offsite-Tuning (OT) (Xiao et al., 2023), constructs a small model by compressing the LLM using layer dropping and knowledge distillation. We match its model size to FedSFT for fair comparison, and use the full Alpaca dataset (52,000 samples) for knowledge distillation because OT requires a large corpus to distill the compressed small model.

**Evaluation Metrics and Protocol.** For the controlled-sentiment generation task, we follow the prior work (Zhou et al., 2024) to utilize the golden reward model Distilbert-imdb[3] to emulate human judgment in classifying movie review sentiments. Distilbert-imdb model assigns a reword score to each response, with higher scores indicating more positive sentiment. For the instruction following task, we use two metrics to ensure a balanced and comprehensive evaluation of the model's ability to generate high-quality, contextually appropriate responses: 1) **Rouge-L score** (Lin, 2004): The Rouge-L score is used to assess the recall and relevance of text generated by a model by measuring the longest common subsequence of words compared to a reference text. Previous works (Wang et al., 2022; Gu et al., 2023) have indicated that Rouge-L is appropriate for large-scale evaluation of instruction-following tasks. 2) **GPT-4 feedback**: We employ GPT-4-Turbo as a judge to evaluate model-generated responses from multiple perspectives, such as helpfulness, relevance, accuracy, and level of detail of their responses. The details are given in Appendix B.3.

In our evaluation process, we extract responses from each model by setting the temperature to 1, limiting responses to a maximum length of 512, and employing random seeds {10, 20, 30, 40, 50}. Following the previous works (Taori et al., 2023; Gu et al., 2023), we utilize a prompt wrapper to reformat each pair of instruction responses into a sentence (See details in Appendix B.4).

## 4.2 CONTROLLED SENTIMENT GENERATION TASK

Figure 4 presents the mean reward scores of FedSFT and baselines on the controlled-sentiment generation task. We highlight four key observations: First, federated fine-tuning via FedIT (120M) yields substantial improvement over the untuned small model (GPT-2 (120M)), though it still lags slightly behind the untuned large model (Base (1.5B)). Second, fine-tuning approaches via surrogates, FedIT+SFT (120M-1.5B) and FedSFT (120M-1.5B), consistently outperform both FedIT (120M) (black dashed line) and the untuned large model GPT-2 (120M) (gray dashed line), achieving reward scores comparable to directly fine-tuning the large model (FedIT (1.5B)). Importantly, these surrogate methods only require fine-

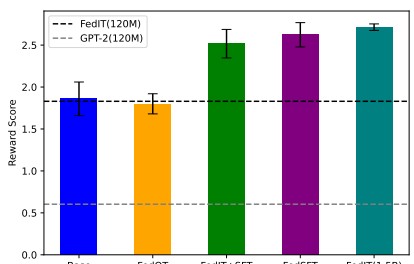

Figure 4: Reward scores of FedSFT and baselines on the controlled-sentiment generation task.

tuning the small model, avoiding the need to train the large model entirely. Third, FedSFT surpasses FedIT+SFT, which we attribute to its use of a large composite model during training for knowledge distillation. This design improves the quality of the aggregated small model. Fourth, FedOT, which uses the first and last two layers as the adapter and two layers as the emulator, performs much worse than FedSFT. This is due to two factors: 1) the emulator requires a large corpus for effective model compression, and 2) OT suffers from a large performance degradation when the layer-drop retention ratio falls below 0.5. These observations align with the findings in Xiao et al. (2023).

---

[3]https://huggingface.co/lvwerra/distilbert-imdb

## 4.3 INSTRUCTION FOLLOWING TASK

The main results are summarized in Table 1, with detailed training outcomes provided in Appendix C.1. Due to GPU memory constraints, distilling an emulator under LLaMA-13B for FedOT is infeasible, and thus FedOT is excluded from this experimental setting.

We draw the following key observations: First, comparing the Base and FedIT methods using large models reveals that: 1) Untuned large models (Base) exhibit suboptimal performance across all instruction-following datasets. 2) Directly fine-tuning large models with FedIT significantly enhances performance on specific downstream tasks. Second, evaluating the methods involving surrogate models yields the following insights: 1) Surrogate fine-tuning can match or even surpass the performance of directly tuned large models. For instance, on the Dolly dataset, although FedIT+SFT (760M-1.5B) and FedSFT (760M-1.5B) only fine-tune GPT-2-large (760M) locally, they achieve Rouge-L scores of 18.9 and 18.6, respectively. These scores surpass the 17.8 achieved by directly federated fine-tuning of GPT-2-large (FedIT (760M)) and are nearly as high as the 19.2 achieved by federated fine-tuning GPT-2-xlarge (FedIT

Table 1: The mean and standard deviation of Rouge-L scores across 5 random seeds for the instruction following task. Larger value indicates better performance.

| Model | Method | Dataset | | |
|---|---|---|---|---|
| | | Dolly | SelfInst | S-NI |
| OPT | Base (13B) | $9.8_{\pm.2}$ | $6.7_{\pm.2}$ | $7.6_{\pm.1}$ |
| | FedIT (13B) | $23.6_{\pm.2}$ | $15.2_{\pm.7}$ | $25.9_{\pm.5}$ |
| | FedIT (1.3B) | $20.4_{\pm.5}$ | $11.5_{\pm.5}$ | $21.0_{\pm.1}$ |
| | FedIT+SFT (1.3B-13B) | $21.5_{\pm.3}$ | $13.0_{\pm.4}$ | $23.3_{\pm.2}$ |
| | FedOT (2-2-2) | $6.1_{\pm.1}$ | $3.7_{\pm.1}$ | $3.8_{\pm.1}$ |
| | FedSFT (1.3B-13B) | $\mathbf{21.8}_{\pm.2}$ | $\mathbf{15.1}_{\pm.6}$ | $\mathbf{25.8}_{\pm.2}$ |
| GPT-2 | Base (1.5B) | $7.2_{\pm.1}$ | $5.5_{\pm.3}$ | $5.8_{\pm.1}$ |
| | FedIT (1.5B) | $19.2_{\pm.4}$ | $11.7_{\pm.7}$ | $22.1_{\pm.4}$ |
| | FedIT (760M) | $17.8_{\pm.5}$ | $10.4_{\pm.3}$ | $18.4_{\pm.3}$ |
| | FedIT+SFT (760M-1.5B) | $18.6_{\pm.4}$ | $10.9_{\pm.5}$ | $21.4_{\pm.3}$ |
| | FedOT (2-18-2) | $5.1_{\pm.2}$ | $4.2_{\pm.1}$ | $4.5_{\pm.1}$ |
| | FedSFT (760M-1.5B) | $\mathbf{18.9}_{\pm.5}$ | $\mathbf{11.0}_{\pm.4}$ | $\mathbf{21.6}_{\pm.2}$ |
| LLaMA | Base (13B) | $9.7_{\pm.2}$ | $7.3_{\pm.5}$ | $8.8_{\pm.1}$ |
| | FedIT (13B) | $24.5_{\pm.3}$ | $19.0_{\pm.8}$ | $29.9_{\pm.5}$ |
| | FedIT (7B) | $23.3_{\pm.6}$ | $17.6_{\pm.6}$ | $25.9_{\pm.2}$ |
| | FedIT+SFT (7B-13B) | $23.5_{\pm.7}$ | $18.9_{\pm.3}$ | $26.7_{\pm.4}$ |
| | FedSFT (7B-13B) | $\mathbf{23.8}_{\pm.4}$ | $\mathbf{19.1}_{\pm.7}$ | $\mathbf{28.7}_{\pm.3}$ |

(1.5B)). 2) FedSFT consistently outperforms FedIT+SFT, as its composite model facilitates knowledge distillation during the training, effectively enhancing the performance of the aggregated small model. 3) Notably, FedSFT (7B-13B) even surpasses FedIT (13B) on the SelfInst dataset, suggesting that surrogate fine-tuning can better preserve essential knowledge than direct fine-tuning, underscoring the strong potential of our approach. Third, consistent with the controlled sentiment generation task, FedOT underperforms on instruction-following datasets as well.

In addition to the Rouge-L results, we report GPT-4 feedback evaluations in Appendix B.3, offering a more holistic assessment of the model's ability to produce helpful, accurate, and contextually appropriate responses. The GPT-4 evaluations exhibit trends consistent with the Rouge-L results.

## 4.4 SYSTEM COST ANALYSIS

The system costs for FedSFT and baselines (with batch size set to 1) are summarized in Table 2. We make the following observations: 1) **Model Size**. Surrogate fine-tuning methods such as FedIT+SFT and FedSFT only require fine-tuning small models on client devices, substantially reducing local storage requirements. 2) **Communication Cost**. By leveraging the small model and LoRA for local fine-tuning, surrogate fine-tuning methods significantly lower communication overhead per client, making them suitable for resource-constrained environments. For example, FedSFT reduces communication cost from 12.5MB to 3MB for OPT models. These savings are especially important in real-world applications where

Table 2: Comparison of system costs. **Model Size** is the model size deployed on clients. **Comm. Cost** is the per-round upload communication cost for each client. **Comp. Cost** is the total TFLOPs for one local iteration. **VRAM** is the peak GPU memory usage.

| Model | Method | Model Size (B) | Comm. Cost (MB) | Comp. Cost (TFLOPs) | VRAM (GB) |
|---|---|---|---|---|---|
| OPT | Base (13B) | N/A | N/A | N/A | N/A |
| | FedIT (13B) | 13.1 | 12.5 | 13.4 | 25.0 |
| | FedIT (1.3B) FedIT+SFT (1.3B-13B) FedSFT (1.3B-13B) | 1.3 | 3.0 | 1.4 | 3.0 |
| GPT-2 | Base (1.5B) | N/A | N/A | N/A | N/A |
| | FedIT (1.5B) | 1.5 | 2.4 | 1.7 | 3.5 |
| | FedIT (760M) FedIT+SFT (760M-1.5B) FedSFT (760M-1.5B) | 0.8 | 1.4 | 0.8 | 2.0 |
| LLaMA | Base (13B) | N/A | N/A | N/A | N/A |
| | FedIT (13B) | 12.9 | 12.5 | 13.4 | 25.8 |
| | FedIT (7B) FedIT+SFT (7B-13B) FedSFT (7B-13B) | 6.7 | 8.1 | 6.9 | 13.5 |

clients may operate over limited-bandwidth or unstable network connections. 3) **Computation**

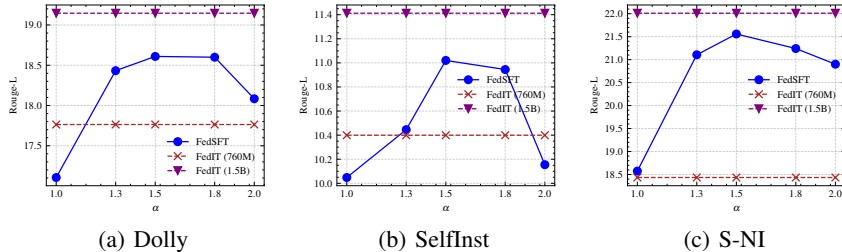

(a) Dolly      (b) SelfInst      (c) S-NI

Figure 5: Performance comparison of FedSFT with different $\alpha$ values on GPT-2 for Dolly, SelfInst, and S-NI datasets. Higher Rouge-L score indicates better performance.

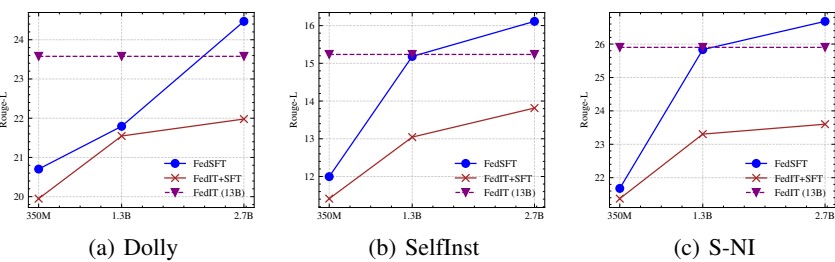

(a) Dolly      (b) SelfInst      (c) S-NI

Figure 6: Performance comparison of FedSFT across different small model sizes on the Dolly, SelfInst, and S-NI datasets. Higher Rouge-L scores indicate better performance.

**Cost.** Surrogate fine-tuning also reduces the computational burden on clients. Since only a small model is tuned with LoRA, the total FLOPs are significantly lower. We measure FLOPs using the DeepSpeed FLOPs profiler[4] (with a batch size of 1 and input length of 512 tokens), and find that FedSFT reduces computational cost by up to a factor of 10 compared to full fine-tuning of large models. 4). **VRAM Usage**. Surrogate fine-tuning methods lower GPU memory requirements, enabling deployment on clients with limited hardware. For instance, FedSFT using OPT-1.3B instead of OPT-13B reduces peak VRAM usage from 25.0GB to just 3.0GB. This makes it feasible for lightweight clients to participate in federated learning without sacrificing model performance.

## 4.5 ABLATION STUDY

**Impact of $\alpha$.** We investigate different values of $\alpha$ in FedSFT to analyze its impact. According to Equation (7), a larger $\alpha$ magnifies the influence of the difference between the tuned and untuned small models, making the predictions more responsive to the fine-tuning adjustments. Conversely, a smaller $\alpha$ results in predictions more similar to the untuned large model. Figure 5 shows results for GPT-2 across three datasets. Notably, for FedSFT on GPT-2, $\alpha = 1.5$ yields the best performance. This demonstrates the importance of carefully tuning $\alpha$ to balance the trade-off between leveraging fine-tuning adjustments and maintaining stability of the untuned large model's predictions.

**Impact of Small Model Scaling.** We fix the LM size to OPT-13B and vary the small model size (OPT-350M, 1.3B, and 2.7B) to evaluate its impact on performance during surrogate fine-tuning. The results are presented in Figure 6. The performance of both FedSFT and FedIT+SFT consistently improves as the small model size increases. Notably, FedSFT can match or even outperform directly federated fine-tuning of LM when using a moderately sized small model in all datasets.

## 5 CONCLUSION

In this paper, we presented FedSFT, a novel FL framework designed to enable efficient fine-tuning of black-box LLMs on resource-constrained clients. FedSFT allows clients to fine-tune LLMs without requiring access to the full model parameters, making it well-suited for practical deployment scenarios. Experiments show that FedSFT achieves performance comparable to direct federated fine-tuning approaches, while significantly reducing computational, communication, and memory overheads. Future work includes analyzing the theoretical convergence of the algorithm.

---

[4]https://www.deepspeed.ai/tutorials/flops-profiler/

ETHICS STATEMENT

All authors have read and agree to the Code of Ethics. This study does not involve any ethical issues.

REPRODUCIBILITY STATEMENT

We have taken several steps to ensure reproducibility. The details of model architectures, hyper-parameters, and evaluation protocols are provided in Appendix B. We also release the code in the anonymous link `https://anonymous.4open.science/r/FedSFT-F808`.

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

APPENDIX

# A MATHEMATICAL DERIVATIONS

## A.1 DERIVATION OF THE OPTIMAL SOLUTION OF THE RL OBJECTIVE

Recall that the fine-tuning can be formulated as the following RL objective:

$$\max_{\mathcal{M}_l^+} \mathbb{E}_{x \sim \mathcal{D}, y \sim \mathcal{M}_l^+(\cdot|x)}[r(x, y) - \beta \mathrm{KL}(\mathcal{M}_l^+(y|x)||\mathcal{M}_l(y|x))], \tag{9}$$

where $r(x, y)$ represents the reward, $\mathcal{M}_l^+$ denotes the large tuned model, $\mathcal{M}_l$ is the large untuned model, and $\beta$ is a parameter that controls the strength of the KL constraint toward $\mathcal{M}_l$. We now have:

$$\max_{\mathcal{M}_l^+} \mathbb{E}_{x \sim \mathcal{D}, y \sim \mathcal{M}_l^+(\cdot|x)}[r(x, y) - \beta \mathrm{KL}(\mathcal{M}_l^+(y|x)||\mathcal{M}_l(y|x))]$$

$$= \max_{\mathcal{M}_l^+} \mathbb{E}_{x \sim \mathcal{D}, y \sim \mathcal{M}_l^+(\cdot|x)}[r(x, y) - \beta \log \frac{\mathcal{M}_l^+(y|x)}{\mathcal{M}_l(y|x)}]$$

$$= \min_{\mathcal{M}_l^+} \mathbb{E}_{x \sim \mathcal{D}, y \sim \mathcal{M}_l^+(\cdot|x)}[\log \frac{\mathcal{M}_l^+(y|x)}{\mathcal{M}_l(y|x)} - \frac{1}{\beta} r(x, y)]$$

$$= \min_{\mathcal{M}_l^+} \mathbb{E}_{x \sim \mathcal{D}, y \sim \mathcal{M}_l^+(\cdot|x)}[\log \frac{\mathcal{M}_l^+(y|x)}{\frac{1}{Z(x)} \mathcal{M}_l(y|x) \exp\left(\frac{1}{\beta} r(x, y)\right)} - \log Z(x)], \tag{10}$$

where we have the partition function:

$$Z(x) = \sum_y \mathcal{M}_l(y|x) \exp\left(\frac{1}{\beta} r(x, y)\right). \tag{11}$$

We can define

$$\tilde{\mathcal{M}}_l(y|x) = \frac{1}{Z(x)} \mathcal{M}_l(y|x) \exp\left(\frac{1}{\beta} r(x, y)\right), \tag{12}$$

which is a valid probability distribution as $\tilde{\mathcal{M}}_l(y|x)$ for all $y$ and $\sum_y \tilde{\mathcal{M}}_l(\cdot|x) = 1$. Since $Z(x)$ is not a function of $y$, we can re-organize the final objective in (10) as:

$$\min_{\mathcal{M}_l^+} \mathbb{E}_{x \sim \mathcal{D}}[\mathbb{E}_{y \sim \mathcal{M}_l^+(\cdot|x)}[\log \frac{\mathcal{M}_l^+(y|x)}{\tilde{\mathcal{M}}_l(y|x)} - \log Z(x)] \tag{13}$$

$$= \min_{\mathcal{M}_l^+} \mathbb{E}_{x \sim \mathcal{D}}[\mathrm{KL}(\mathcal{M}_l^+(y|x), \tilde{\mathcal{M}}_l(y|x)) - \log Z(x)]. \tag{14}$$

Since $Z(x)$ is independent of the $\mathcal{M}_l^+$, minimizing the objective reduces to minimizing the KL divergence term. By Gibbs' inequality, the KL divergence achieves its minimum value of zero if and only if the two distributions are identical. Therefore, the optimal policy is obtained by matching $\tilde{\mathcal{M}}_l(y|x)$ to the normalized reference distribution:

$$\tilde{\mathcal{M}}_l(y|x) = \frac{1}{Z(x)} \mathcal{M}_l(y|x) \exp\left(\frac{1}{\beta} r(x, y)\right). \tag{15}$$

## A.2 PROOF OF THEOREM 1

*Proof.* In our method, the large model $\mathcal{M}_l$ is a black box and cannot be directly fine-tuned. Instead, we fine-tune the small model $\mathcal{M}_s$ to obtain $\mathcal{M}_s^+$, and define the reward as $r(x, y) = \beta \log \frac{\mathcal{M}_s^+(y|x)}{\mathcal{M}_s(y|x)}$. According to (15), the surrogate fine-tuned large model is then constructed by

$$\tilde{\mathcal{M}}_l(y|x) = \frac{1}{Z(x)} \mathcal{M}_l(y|x) \frac{\mathcal{M}_s^+(y|x)}{\mathcal{M}_s(y|x)} \propto \mathcal{M}_l(y|x) \frac{\mathcal{M}_s^+(y|x)}{\mathcal{M}_s(y|x)}. \tag{16}$$

Then, we have:

$$\log \tilde{\mathcal{M}}_l(y|x)) = \log \mathcal{M}_l(y|x) + \log \mathcal{M}_s^+(y|x) - \log \mathcal{M}_s(y|x). \tag{17}$$

Inspired by (17), we design a composite teacher model that directly implements this decomposition. Specifically, we use the logits from the large untuned model $g_{\mathcal{M}_l}$ and the difference in logits between the tuned and untuned small model $(g_{\mathcal{M}_s^+} - g_{\mathcal{M}_s})$ to represent the task-specific reward signal. Then, the probability distribution of the next token prediction from the composite model $\tilde{\mathcal{M}}_l$ is given by

$$\pi_{\tilde{\mathcal{M}}_l}(X_t|x_{<t}) := \text{softmax}\left[g_{\mathcal{M}_l}(X_t|x_{<t}) + \alpha\left(g_{\mathcal{M}_s^+}(X_t|x_{<t}) - g_{\mathcal{M}_s}(X_t|x_{<t})\right)\right], \qquad (18)$$

where $g_{\mathcal{M}_l}$, $g_{\mathcal{M}_s^+}$, and $g_{\mathcal{M}_s}$ are the logit scores over the vocabulary produced by the language modeling heads of models $\mathcal{M}_l$, $\mathcal{M}_s^+$, and $\mathcal{M}_s$, respectively. The hyperparameter $\alpha$ controls the influence on the output distribution of the $\mathcal{M}_l$. A smaller value of $\alpha$ results in predictions that closely resemble those of the $\mathcal{M}_l$, whereas a larger $\alpha$ magnifies the contrast between $\mathcal{M}_s^+$ and $\mathcal{M}_s$. $\qquad\qquad\square$

# B EXPERIMENT DETAILS

## B.1 DATASET

"imdb-preference" dataset [5] consists of 20,000 training data points, each containing a prompt and a pair of responses scored by a golden reward model. We select the higher-scored positive response to encourage positive continuation of movie reviews. We randomly sample 128 data points for knowledge distillation and allocate the remaining data across 10 clients for training.

Dolly ("databricks/databricks-dolly-15k") [6] is an open-source collection of 15,000 high-quality human-generated prompt and response pairs designed for training and evaluating natural language processing models. We remove samples that surpass the models' context length. Then, we randomly allocate 1,000 samples for validation and 500 for testing, thereby retaining approximately 12,500 examples dedicated to training purposes. This dataset covers a range of instructional categories, including brainstorming, classification, closed-question answering (QA), generation, information extraction, open QA, and summarization. These categories were chosen to reflect different types of cognitive tasks that could be useful for training LLMs to respond in human-like manners across a variety of contexts. We plot the number of data samples and their corresponding percentage in Figure 7.

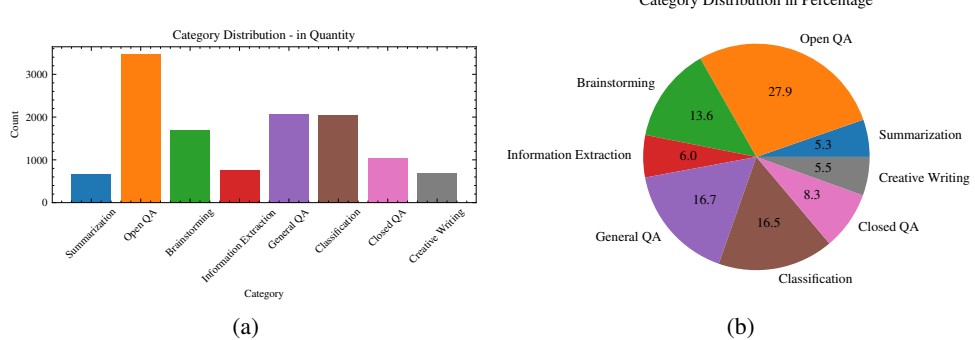

(a)                                                   (b)

Figure 7: Bar and pie charts of the number (a) and corresponding percentage (b) of each category in the Dolly dataset.

SelfInst dataset [7] is designed to evaluate the practical utility of instruction-following models in user-oriented contexts. This dataset includes a diverse array of tasks accompanied by specific instructions, including tables, codes, or math equations. In total, it contains 252 distinct tasks, each associated with a unique instruction, aimed at testing the capability of models across a broad spectrum of applications. We show the number of data samples in the test dataset from each category in Figure 8.

---

[5] https://huggingface.co/datasets/ZHZisZZ/imdb_preference
[6] https://huggingface.co/datasets/databricks/databricks-dolly-15k
[7] https://github.com/yizhongw/self-instruct

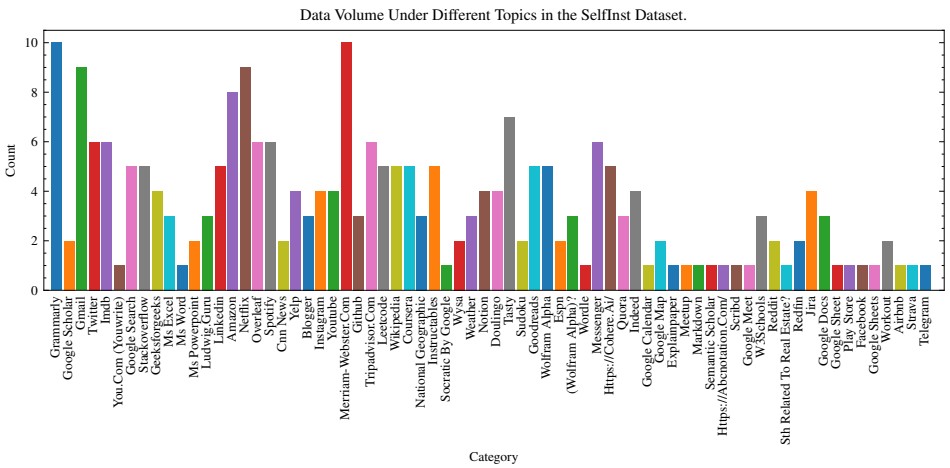

Figure 8: Bar chart of the number of each category in the SelfInst dataset.

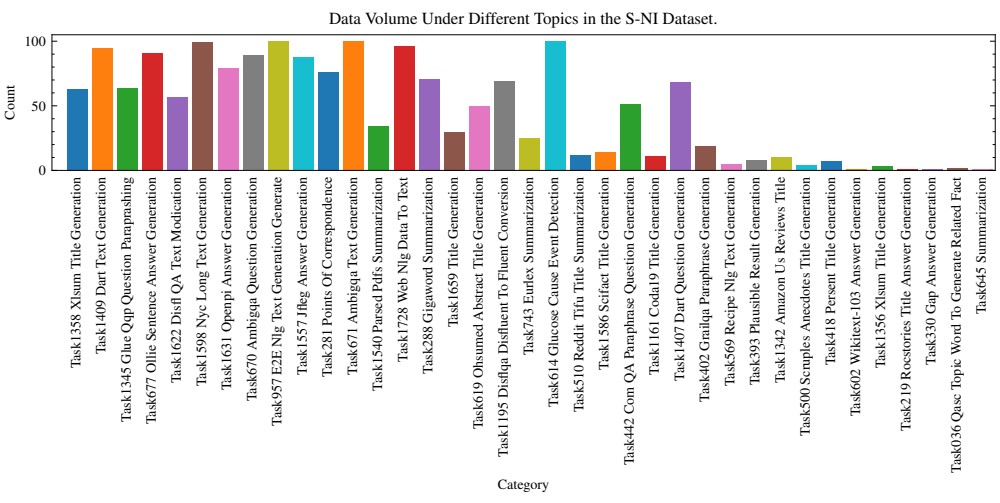

Figure 9: Bar chart of the number of each category in the S-NI dataset.

S-NI dataset (Super-NaturalInstructions) (Wang et al., 2022) is designed to test the generalization capabilities of large models across a wide range of NLP tasks through declarative instructions. It includes over 1,600 unique tasks, encompassing diverse categories such as text classification, summarization, question answering, and more complex reasoning tasks. We draw the number of data samples used for evaluation in each category in Figure 9.

## B.2 DATA PARTITION STRATEGY

Federated fine-tuning LLMs involves tuning algorithms across multiple decentralized devices or servers holding local data samples, which are usually not identically distributed. This scenario frequently occurs in real-world applications, where data naturally varies across clients due to geographic diversity and user behavior. For example, diverse clients might engage in distinct activities like open-domain QA and creative writing. In this case, the format and content of instructions can be significantly different. For instance, QA tasks often focus on factual queries and responses, whereas creative writing tasks require guidelines for crafting engaging and imaginative narratives.

To simulate an FL setup, we employ two data partition strategies, pathological non-IID (McMahan et al., 2017) and Dirichlet non-IID (Hsu et al., 2019). Specifically, we first sort the data from the Dolly dataset by categories. Then we randomly partition the dataset into 10 shards. For pathological

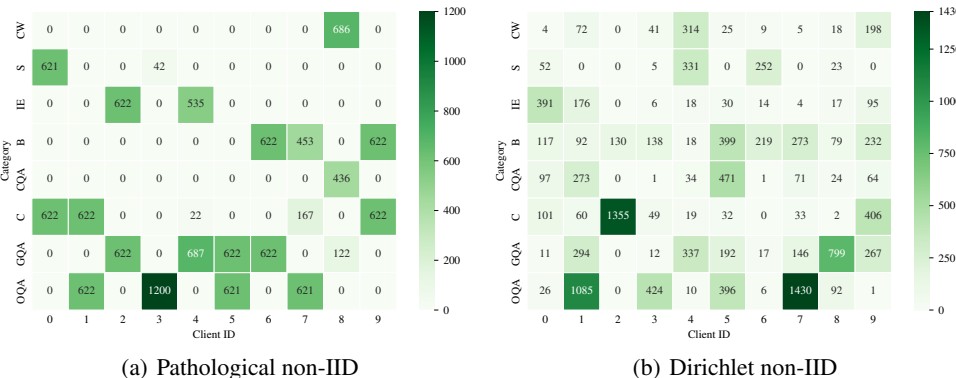

(a) Pathological non-IID        (b) Dirichlet non-IID

Figure 10: (a) The pathological non-IID distribution of instruction categories distribution across clients. Categories: creative writing(CW), summarization(S), information extraction(IE), brainstorming(B), closed QA(CQA), classification(C), general QA(GQA), open QA(OQA). (b) The Dirichlet non-IID distribution of instruction categories across clients.

Table 3: Evaluation results on Dirichlet distribution. We report the average and standard deviation of Rouge-L scores across 5 random seeds. Higher values indicate better performance.

| Model | Method | Dataset | | |
|---|---|---|---|---|
| | | Dolly | SelfInst | S-NI |
| | FedIT (1.5B) | $19.1_{\pm.6}$ | $11.2_{\pm.4}$ | $20.7_{\pm.3}$ |
| GPT-2 | Base (1.5B) | $7.2_{\pm.1}$ | $5.5_{\pm.3}$ | $5.8_{\pm.1}$ |
| | FedIT (760M) | $18.0_{\pm.5}$ | $10.1_{\pm1.}$ | $17.1_{\pm.3}$ |
| | FedIT+SFT (760M-1.5B) | $18.6_{\pm.5}$ | $10.2_{\pm.8}$ | $19.7_{\pm.3}$ |
| | FedSFT (760M-1.5B) | $\mathbf{18.8}_{\pm.4}$ | $\mathbf{11.2}_{\pm.4}$ | $\mathbf{20.3}_{\pm.2}$ |

non-IID distribution, each shard contains an equal number of samples and exclusively represents two specific categories. For the Dirichlet distribution, the data from the same category are distributed among shards following the Dirichlet distribution with concentration parameter 0.5. These segmentation strategies followed a commonly used partitioning method in (Zhang et al., 2024; He et al., 2020; Lai et al., 2022; Zhang et al., 2023), which led to a non-IID data distribution among the clients with imbalanced categories of instructions, mirroring a typical real-world FL data distribution. Figure 10 depicts the distribution of instruction categories within each client's dataset, with the former showing the pathological distribution and the latter displaying the Dirichlet distribution, respectively. As shown in Figure 10, for pathological distribution, each client has imbalanced instruction categories with some categories completely missing. For the Dirichlet distribution, Figure 10 illustrates that each client has an imbalanced distribution of instruction categories and a varying total number of samples. These imbalances mirror real-world conditions, where individual users often encounter a skewed variety of instructions, reflecting their unique usage patterns and preferences.

We apply the Dirichlet distribution to the training dataset and present the evaluation results for the GPT-2 models in Table 3. The results indicate that FedSFT outperforms the federated fine-tuning small model, such as FedIT (760M) and FedIT+SFT (760M-1.5B), and performs comparably to federated fine-tuning large models like FedIT (1.5B).

### B.3 GPT-4 EVALUATION

We use the Rouge-L scores and GPT-4 feedback scores to evaluate the model-generated responses. These approaches ensure a more balanced and comprehensive evaluation of the model's ability to produce high-quality, contextually appropriate text. Following the same evaluation approach in (Gu et al., 2023; Shen et al., 2023), we utilize GPT-4 as a judge to compare model-generated responses

```
[Instruction]
{instruction}
[Input]
{input}
[The Start of Assistant 1's response]
{answer 1}
[The End of Assistant 1's Answer]
[The Start of Assistant 2's response]
{answer 2}
[The End of Assistant 2's Answer]
[System]
We would like to request your feedback on the performance of two AI
assistants in response to the user instruction and input displayed above.
Please rate the helpfulness, relevance, accuracy, and level of detail of
their responses. Each assistant receives an overall score on a scale of 1
to 10, where a higher score indicates better overall performance.
Please first provide a comprehensive explanation of your evaluation,
avoiding any potential bias and ensuring that the order in which the re-
sponses were presented does not affect your judgment.
Then, output two lines indicating the scores for Assistant 1 and 2, re-
spectively.
Output with the following format:
Evaluation evidence: <your evaluation explanation here>
Score of the Assistant 1: <score>
Score of the Assistant 2: <score>
```

Figure 11: GPT-4 evaluation prompt.

with the ground truth answers, assigning scores from 1 to 10 for both sets of responses. We call the GPT-4 Turbo API[8] with the temperature $= 0.7$. The evaluation prompt used for GPT-4 is illustrated in Figure 11. We calculate the ratio of the total scores of model-generated responses and the ground truth answers. We select the seed closest to the average Rouge-L score and then report its GPT-4 feedback score. For the Dolly and SelfInst datasets, we evaluate all the responses. For the S-NI dataset, we randomly select 200 responses for evaluation. The results are summarized in Table 4. Due to the poor performance of the base untuned large models and FedOT, we have excluded their GPT-4 scores in Table 4. These tables demonstrate that FedSFT can achieve performance compara-ble to the direct tuning of the large models in the FL setting.

### B.4 AUTOMATIC EVALUATION DETAILS

In our evaluation process, we extract responses from each model by setting the temperature to 1, limiting responses to a maximum length of 512, and employing random seeds [10, 20, 30, 40, 50]. Following the previous works Taori et al. (2023); Gu et al. (2023), we utilize a prompt wrapper illustrated in Figure 12 to reformat each pair of instruction-response into a sentence.

### B.5 HYPERPARAMETERS

The specific configurations are documented in Table 5. For all experiments, we use the most com-mon PEFT technique, LoRA (Hu et al., 2021), for our local training. We fine-tune the models for 20 communication rounds using the Prodigy optimizer (Mishchenko & Defazio, 2024), with an initial learning rate of 1. A cosine learning rate decay strategy (Loshchilov & Hutter, 2016) is applied at each communication round, and safeguard warmup without bias correction is implemented. To save the memory footprint, all models are loaded into VRAM in half-precision mode, with checkpoints also saved in this format. For knowledge distillation of FedSFT, the hyperparameter $\lambda$ is set to 0.1.

---

[8]API version of 2024-04-09.

Table 4: Evaluation results by GPT-4 feedback. Higher scores indicate better performance.

| Model | Method | Dataset | | |
|---|---|---|---|---|
| | | Dolly | SelfInst | S-NI |
| OPT | FedIT (13B) | 51.9 | 44.5 | 44.9 |
| | FedIT (1.3B) | 37.0 | 28.4 | 29.2 |
| | FedIT+SFT (1.3B-13B) | 43.4 | 35.4 | 35.3 |
| | FedSFT (1.3B-13B) | **44.3** | **42.6** | **41.1** |
| GPT-2 | FedIT (1.5B) | 35.7 | 29.1 | 29.2 |
| | FedIT (760M) | 30.3 | 26.1 | 24.5 |
| | FedIT+SFT (760M-1.5B) | 34.4 | 28.2 | 26.8 |
| | FedSFT (760M-1.5B) | **34.8** | **28.8** | **27.8** |
| LLaMA | FedIT (13B) | 65.4 | 59.5 | 61.8 |
| | FedIT (7B) | 57.7 | 52.1 | 50.7 |
| | FedIT+SFT (7B-13B) | 63.6 | 56.4 | 59.0 |
| | FedSFT (7B-13B) | **65.4** | **60.3** | **61.6** |

---

Below is an instruction that describes a task.
Write a response that appropriately completes the request.

[Instruction]
{instruction}

[Input]
{input}

[Response]

---

Figure 12: The prompt wrapper for training and evaluation.

During evaluation, we consistently generate responses using greedy search with unrestricted sampling. The Top-p ratio is set to 1.0 and the temperature to 1.0. The maximum generation length is capped at 512 tokens. Evaluation batch sizes are 32 for the GPT-2 model and 8 for the OPT/LLaMA model, respectively.

## B.6 LoRA CONFIGURATION

We apply LoRA to the attention layer for GPT-2 model and "q_proj", "v_proj" layers for LLaMA model to enhance adaptation capabilities, using the Adam optimizer for effective training. We set the rank of LoRA to be 4 and 8 for GPT-2 and LLaMA, respectively. This only yields 4.2 M trainable parameters with size 8.1 MB for LLaMA-7B model, which is affordable for many user devices. The overall LoRA training configuration for different models can be found in Table 6.

## B.7 HARDWARE AND LIBRARY

We conduct the experiment on the Ubuntu (22.04.4 LTS) server equipped with 4 A6000 GPUs. Each GPU has 48 GB VRAM. The training scripts were implemented using Pytorch 2.0.1 (Paszke et al., 2019). To accelerate the experiment's progress, we also employ popular open-sourced third-party packages, including transformer 4.36.0.dev0 (Wolf et al., 2020), deepspeed 0.14.0 (Rasley et al., 2020), accelerate 0.29.2 (Gugger et al., 2022), nltk 3.8.1 (Bird et al., 2009), sentencepiece 0.2.0 (Kudo & Richardson, 2018), and datasets 2.81.0 (Lhoest et al., 2021). For LoRA local training,

Table 5: Hyperparameters for fine-tuning models.

| Hyperparameter | GPT-2 | OPT | LLaMA |
|---|---|---|---|
| Precision | Float16 | Float16 | Float16 |
| Number of local epochs | 2 | 2 | 2 |
| Total round | 20 | 20 | 20 |
| Training Batch size | 64 | 64 | 64 |
| LoRA rank | 4 | 8 | 8 |
| Weight decay | 0.01 | 0.01 | 0.01 |
| Max sequence length | 512 | 512 | 512 |
| KD data size | 128 | 512 | 512 |
| KD batch size | 16 | 32 | 32 |
| KD iterations | 8 | 16 | 16 |

Table 6: LoRA training configuration for fine-tuning models.

| Model | #Size | Rank | Trainable Param | LoRA Size | Trainable Fraction |
|---|---|---|---|---|---|
| GPT-2 | 760M | 4 | 0.7 M | 1.4 MB | 0.09% |
| | 1.5B | 4 | 1.2 M | 2.4 MB | 0.08% |
| OPT | 350M | 8 | 0.8 M | 3.0 MB | 0.24% |
| | 1.3B | 8 | 1.6 M | 6.0 MB | 0.12% |
| | 2.7B | 8 | 2.6 M | 10.0 MB | 0.10% |
| | 13B | 8 | 6.6 M | 25.0 MB | 0.05% |
| LLaMA | 7B | 8 | 4.2 M | 8.1 MB | 0.06% |
| | 13B | 8 | 6.5 M | 12.5 MB | 0.05% |
| | 30B | 8 | 12.8 M | 25.6 MB | 0.04% |

we implement the low-rank model update using PEFT package (Mangrulkar et al., 2022). For all experiments, we adopt the Python 3.10 interpreter and CUDA version 11.4.

## C  RESULT ANALYSIS

### C.1  DETAILED TRAINING PROCESS

In this section, we conduct a comprehensive training process comparison of FedSFT and the baselines across Dolly, SelfInst, and S-NI datasets for the instruction following task. The detailed results during training are depicted in Figure 13.

### C.2  EVALUATION OF TOKENS MOST INFLUENCED BY SURROGATE FINE-TUNING

We aim to investigate which tokens are most influenced by FedSFT. To this end, we calculate the frequency of each token in the generated responses for GPT-2-1.5B to its FedSFT version. Table 7 summarizes the 8 tokens whose occurrence frequency has increased from GPT-2-1.5B to its FedSFT. We can see that these tokens are more contributing to reasoning and style. These findings are consistent with the hypothesis that instruction-tuning mainly influences reasoning and style, rather than increasing the model's knowledge (Gudibande et al., 2023).

### C.3  GENERATION DIVERSITY

Table 7 shows that the occur frequency of certain tokens increased from LLaMA-7B to its surrogate-tuned version, potentially affecting generation diversity. To investigate this impact, we conducted experiments on distinct n-grams (Dist-3 and Dist-4) diversity, a widely used metric to measure the

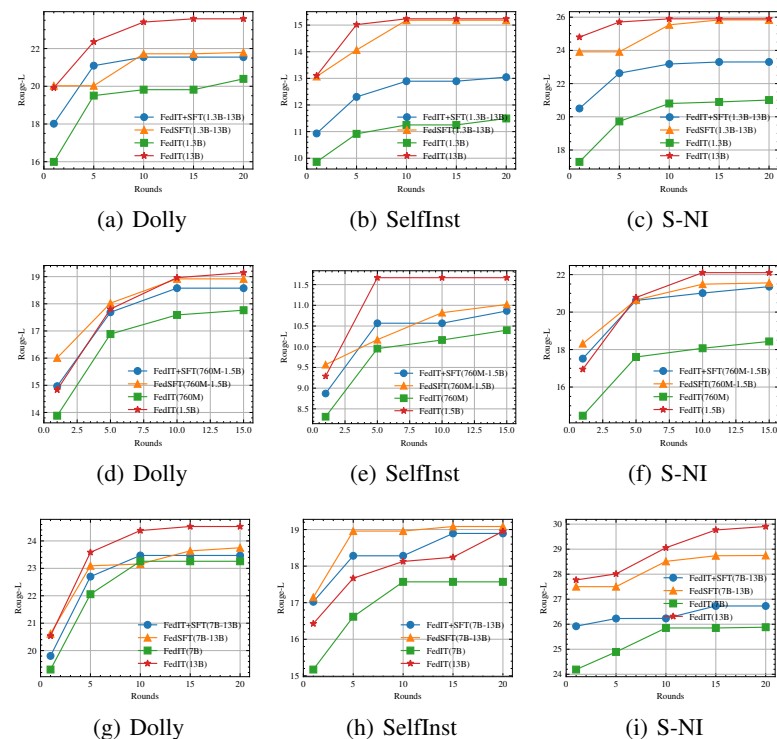

Figure 13: Evaluation results of FedSFT and baselines on OPT (a, b, c), GPT-2 (d, e, f), and LLaMA (g, h, i) models across different rounds for Dolly, SelfInst, and S-NI datasets. FedOT is omitted as it fails to converge. Higher Rouge-L scores indicate better performance.

Table 7: For the three datasets, the 8 tokens whose occurrence frequency increased the most from GPT-2-1.5B to its FedSFT version. Top Context shows the most common 3-gram that the word occurs in.

| Dolly | | SelfInst | | S-NI | |
|---|---|---|---|---|---|
| Token | Top Context | Token | Top Context | Token | Top Context |
| is | is one of the | equal | is equal to | because | because he is |
| a | it can be a | well | as well as | he | when he was |
| can | can be used to | was | said it was | in | facts specified in |
| popular | the most popular | do | need to do | they | they are not |
| most | of the most | it | and it will | changed | entity changed from |
| known | is known for | into | into something new | has | an individual has |
| when | when I was | nothing | there is nothing | were | and I were |
| many | there are many | when | when a change | is | there is a |

generation diversity of LLMs (Li et al., 2016) (see Appendix C.4 for more details). As shown in Table 8, our algorithm maintains a high level of diversity despite the observed changes in token frequency.

## C.4 DETAILS ABOUT GENERATION DIVERSITY METRICS

Dist-n is calculated as a fraction $N/C$, where $N$ represents the number of distinct n-grams in the generated responses and $C$ denotes the total number of generated n-grams. We report the average values across 5 seeds in Table 8.

Table 8: The distinct 3-grams and 4-grams (Dist-3 and Dist-4) on the test sets. FedSFT preserves generation diversity.

| Model | Method | Dolly | | SelfInst | | S-NI | |
|---|---|---|---|---|---|---|---|
| | | Dist-3 | Dist-4 | Dist-3 | Dist-4 | Dist-3 | Dist-4 |
| LLaMA | FedIT (13B) | 96.4 | 99.3 | 97.5 | 99.4 | 93.2 | 98.0 |
| | Base (13B) | 95.0 | 99.1 | 96.3 | 99.3 | 89.9 | 97.9 |
| | FedIT (7B) | 96.6 | 99.3 | 97.7 | 99.5 | 93.5 | 98.4 |
| | FedIT+SFT | 97.1 | 99.4 | 98.0 | 99.6 | 94.3 | 98.5 |
| | FedSFT | 97.2 | 99.4 | 98.0 | 99.5 | 93.8 | 98.2 |
| GPT-2 | FedIT (1.5B) | 97.0 | 99.4 | 98.1 | 99.5 | 94.7 | 98.5 |
| | Base (1.5B) | 96.3 | 99.4 | 97.0 | 99.4 | 92.2 | 95.6 |
| | FedIT (760MB) | 97.0 | 99.4 | 98.2 | 99.6 | 94.8 | 98.6 |
| | FedIT+SFT (760M-1.5B) | 97.0 | 99.4 | 98.3 | 99.6 | 94.7 | 98.3 |
| | FedSFT (760M-1.5B) | 97.4 | 99.5 | 98.5 | 99.6 | 93.3 | 97.5 |

## C.5 FURTHER EVALUATION OF $\alpha$

We use different $\alpha$ values in FedSFT to investigate the effect of surrogate-tuning weight $\alpha$. Specifically, we set $\alpha \in \{1.0, 1.3, 1.5, 1.8, 2.0\}$ to evaluate the GPT-2 model on three testing datasets at global rounds $\{1, 5, 10, 15\}$. For the LLaMA model, we evaluate it at global rounds $\{1, 5, 10, 15, 20\}$, using $\alpha$ values in the range $\{1.0, 1.5, 2.0\}$. The Rouge-L scores for LLaMA and GPT-2 are illustrated in Figure 14. From these figures, we can find that the value $\alpha$ plays a crucial role in determining the behavior of the model. As $\alpha$ increases, the influence of the tuned small model on the predictions becomes more pronounced, leading to more substantial deviations from the un-tuned large model's behavior. Conversely, as $\alpha$ decreases, the predictions tend to align more closely with the target untuned large model, resulting in a more stable and conservative output. Therefore, in practice, we need to carefully choose an appropriate $\alpha$ for the specific downstream task.

## C.6 IMPACT OF LARGE MODEL SCALING

We investigate the performance of FedSFT and FedIT+SFT when we scale up the size of a large model. Figure 15 shows the best Rouge-L score, while the training dynamics are summarized in Figure 16. Specifically, we reuse the tuned LLaMA-7B from FedSFT (7B-30B) to surrogate-tune LLaMA-30B. Similarly, we evaluate FedIT+SFT on LLaMA-30B, reusing the tuned LLaMA-7B from FedIT (7B). This approach is designed to simulate a realistic scenario in which, during the training phase, only LLaMA-7B and LLaMA-13B are used. In the deployment phase, however, if more powerful models such as LLaMA-30B become available, we aim to evaluate whether the model trained with FedSFT retains its advantage over FedIT+SFT. From Figure 15, we observe that performance improves for both FedSFT and FedIT+SFT as the large model size increases. The results for both methods show a clear positive correlation between model size and performance, highlighting the scaling law: larger models yield better results. Additionally, FedSFT consistently outperforms FedIT+SFT, reinforcing the advantage of our proposed method as the number of model parameters increases.

## C.7 EFFECTIVENESS UNDER LIMITED LOGITS

In practice, closed-source LLM APIs often restrict logit access and return only top-$k$ probabilities for each generated token. To evaluate whether FedSFT remains effective under such constraints, we consider an extreme setting in which the large model provides *only its top-5 logit outputs* instead of the full logits. As shown in Table 9, FedSFT maintains strong performance even when the large model exposes only top-5 logits, achieving results close to those obtained with full logits. This demonstrates that FedSFT does *not* rely on access to complete logit vectors, and is robust to the limited logit access commonly found in real-world API-based LLM deployments.

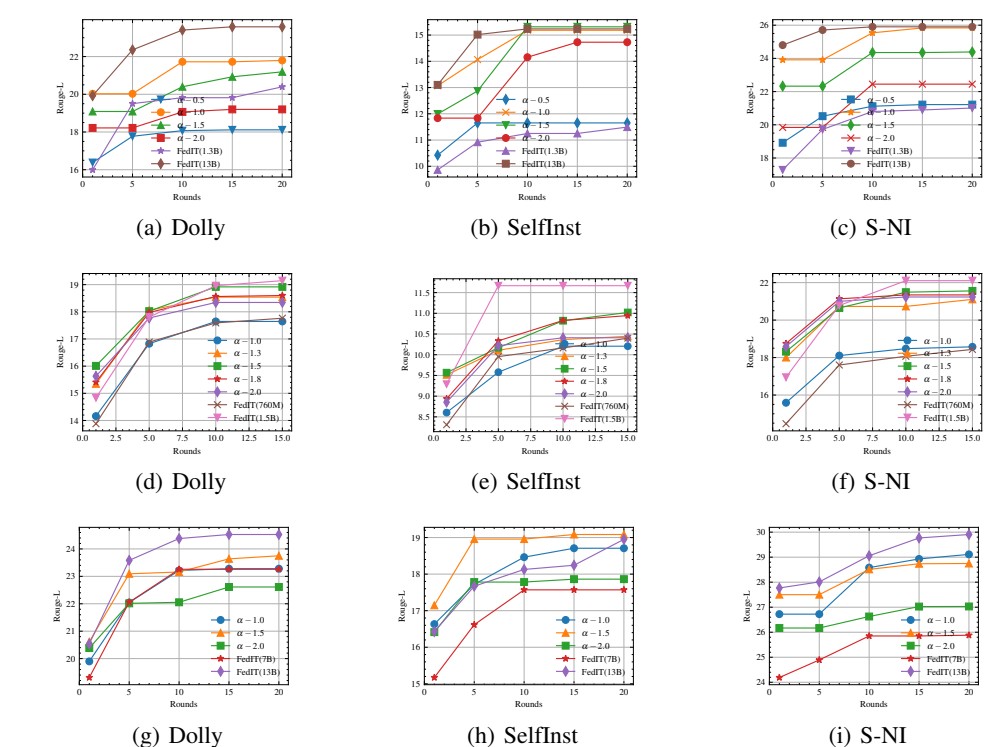

Figure 14: Performance comparison of different $\alpha$ for FedSFT on OPT (a, b, c), GPT-2 (d, e, f), and LLaMA (g, h, i) across different rounds for Dolly, SelfInst, and S-NI tasks. Higher Rouge-L scores indicate better performance.

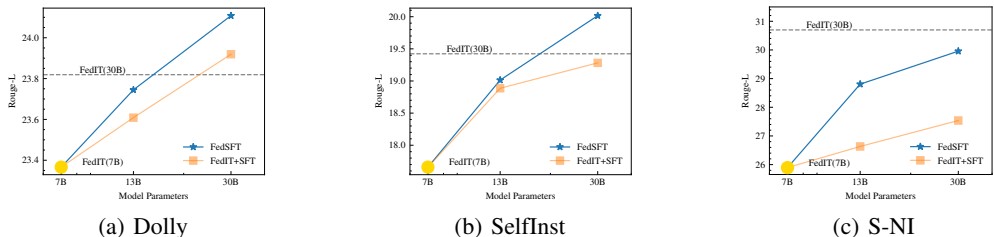

Figure 15: The scaling of large model models in the LLaMA family. FedIT (7B) and FedIT (30B) are directly tuned models by FedIT. At the 13B scale, we report the performance of FedSFT (7B-13B) and FedIT+SFT (7B-13B). At the 30B scale, we use the tuned 7B model from FedSFT (7B-13B) to surrogate fine-tune the 30B model for FedSFT, and the 7B model from FedIT (7B) to surrogate fine-tune the 30B model for FedIT+SFT.

## C.8    IMPACT OF AGGREGATION

To complement our ablation analysis and isolate the contribution of aggregation in FedSFT, we introduce two additional baselines. 1) CentralSFT, which performs centralized fine-tuning on the small model followed by distillation using a composite teacher that includes the large model. 2) LocalSFT, where each client fine-tunes its small model independently and performs composite model distillation with the server without any aggregation. Together, CentralSFT and LocalSFT provide meaningful upper- and lower-bound references for understanding the impact of aggregation in FedSFT.. As shown in Table 10, FedSFT consistently outperforms the local-only baseline (LocalSFT) and approaches the centralized baseline (CentralSFT) across all datasets.

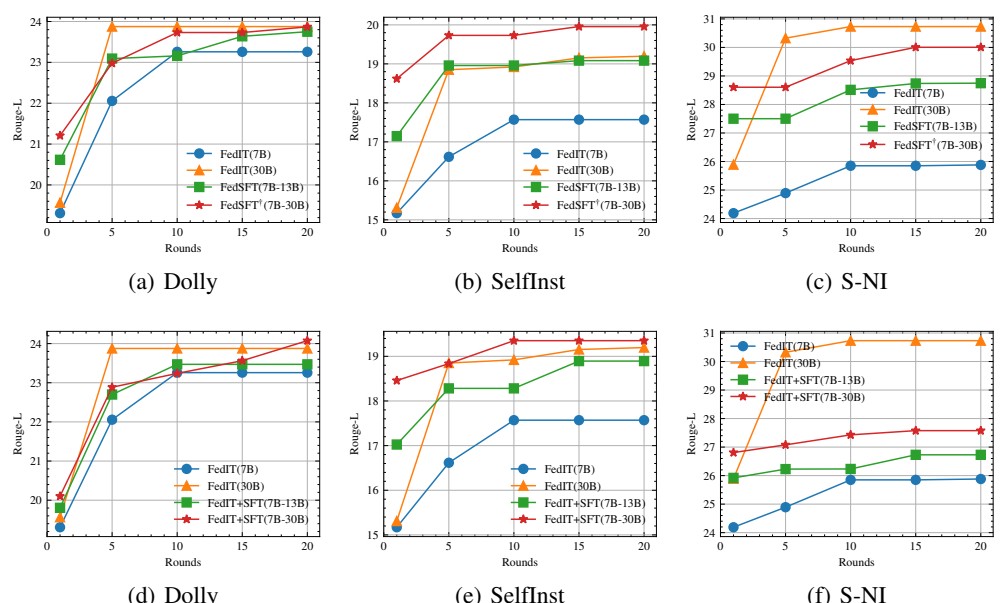

Figure 16: The scaling of large models in the LLaMA family models. (a, b, c) are the results of FedSFT. (d, e, f) are the results of FedIT+SFT. FedSFT† (7B-30B) uses the tuned 7B model from FedSFT (7B-13B) to surrogate fine-tune the LLaMA 30B.

Table 9: The mean and standard deviation of Rouge-L scores across 5 random seeds with top-5 logits from the large model for the instruction following task. A larger value indicates better performance.

| Model | Method | Dataset | | |
|---|---|---|---|---|
| | | Dolly | SelfInst | S-NI |
| OPT | Base (13B) | $9.8_{\pm.2}$ | $6.7_{\pm.2}$ | $7.6_{\pm.1}$ |
| | FedIT (13B) | $23.6_{\pm.2}$ | $15.2_{\pm.7}$ | $25.9_{\pm.5}$ |
| | FedIT (1.3B) | $20.4_{\pm.5}$ | $11.5_{\pm.5}$ | $21.0_{\pm.1}$ |
| | FedIT+SFT (top-5) (1.3B-13B) | $23.4_{\pm.3}$ | $14.7_{\pm.4}$ | $30.7_{\pm.2}$ |
| | FedOT (2-2-2) | $6.1_{\pm.1}$ | $3.7_{\pm.1}$ | $3.8_{\pm.1}$ |
| | FedSFT (top-5) (1.3B-13B) | $\mathbf{23.7}_{\pm.3}$ | $\mathbf{15.3}_{\pm.3}$ | $\mathbf{31.6}_{\pm.3}$ |

## C.9 PERFORMANCE-EFFICIENCY TRADE-OFF ANALYSIS

To provide a unified view of the performance-efficiency trade-off, we report the ROUGE-L scores and system costs of FedSFT and FedIT+SFT under different client model sizes. Table 11 summarizes the results across three datasets (Dolly, SelfInst, and S-NI) along with communication cost, computation cost, and VRAM usage.

We observe a clear pattern: larger client models achieve higher downstream performance but incur proportionally larger computation, communication, and memory overhead. More importantly, FedSFT consistently attains higher accuracy than FedIT+SFT under the same resource budget, demonstrating a better accuracy–efficiency balance.

Table 10: Rouge-L scores across 5 random seeds. A larger value indicates better performance.

| Model | Method | Dataset | | |
|---|---|---|---|---|
| | | Dolly | SelfInst | S-NI |
| OPT | LocalSFT (1.3B-13B) | $20.7_{\pm 1.1}$ | $11.9_{\pm .8}$ | $20.9_{\pm 2.1}$ |
| | CentralSFT (1.3B-13B) | $23.7_{\pm .3}$ | $15.5_{\pm .8}$ | $26.1_{\pm .2}$ |
| | FedSFT (1.3B-13B) | $21.8_{\pm .2}$ | $15.1_{\pm .6}$ | $25.8_{\pm .2}$ |

Table 11: Performance-efficiency trade-off under different client model sizes for FedSFT. We report ROUGE-L scores on Dolly, SelfInst, and S-NI. The server model is fixed at 13B.

| Method | Size | Dolly | SelfInst | S-NI | Comm. Cost (MB) | Comp. Cost (TFLOPs) | VRAM (GB) |
|---|---|---|---|---|---|---|---|
| FedSFT | 350M | 20.7 | 12.0 | 21.7 | 1.4 | 0.4 | 0.9 |
| | 1.3B | 21.8 | 15.2 | 25.8 | 3.0 | 1.4 | 3.0 |
| | 2.7B | 24.5 | 16.1 | 26.7 | 5.1 | 2.8 | 5.8 |
| FedIT+SFT | 350M | 19.9 | 11.4 | 21.4 | 1.4 | 0.4 | 0.9 |
| | 1.3B | 21.5 | 13.0 | 23.3 | 3.0 | 1.4 | 3.0 |
| | 2.7B | 22.0 | 13.8 | 23.6 | 5.1 | 2.8 | 5.8 |

## D  SUPPORTING PLOTS

In this section, we analyze the Rouge-L score and BLEU score among different category distributions for different models tuned by FedSFT. Here, we only show the result of GPT-2 (760M-1.5B) and LLaMA (7B-13B).

### D.1  PLOTS FOR CATEGORY SCORES OF DOLLY

In Figure 17, 18, we plot the Rouge-L score and BLEU score from different categories of the Dolly dataset at round 1, 15 on GPT-2 model and round 1, 20 on the LLaMA model. From the figures, we can find the scores for most categories are continually improving as the global rounds increase. The category "classification" leads the most contribution during training. In global round 1, the performance across tasks is fairly uniform, hovering around an overall average Rouge-L score (indicated by the dashed line), with "classification" scoring notably higher. By global rounds 15 and 20, there is a clear shift in performance; "classification" peaks significantly above other tasks, suggesting an improvement in the system's capability to handle classification tasks, while the other tasks show varied but generally less substantial improvement. The error bars for Rounds 15 and 20 tend to be smaller across various tasks, suggesting a potential decrease in variability and enhanced consistency in the model's performance among different random seeds.

### D.2  PLOTS FOR CATEGORY SCORES OF SELFINST

In Figure 19,20, we plot the Rouge-L score and BLEU score from different categories on the Self-Inst dataset at round 1, 15 on GPT-2 model and round 1, 20 on LLaMA model. From the figures, we can find the tasks evaluated cover a broad range of services, from search engines like Google, and social media platforms like Instagram and Twitter, to productivity tools like Microsoft Word and Google Sheets. Notably, some tasks like "Google Sheet" and "Markdown" score particularly high, suggesting that text generated or retrieved in these contexts has a high degree of fidelity to the expected reference texts. Conversely, tasks involving more dynamic or personalized content, such as "Twitter," "Facebook," and "YouTube" show lower scores, which could be due to the more challenging nature of predicting or matching varied user-generated content. The graph also highlights specific domains that involve deeper domain knowledge, such as Leetcode, Quora, and Reddit, where the Rouge-L scores fall below the overall average. This observation suggests that the model

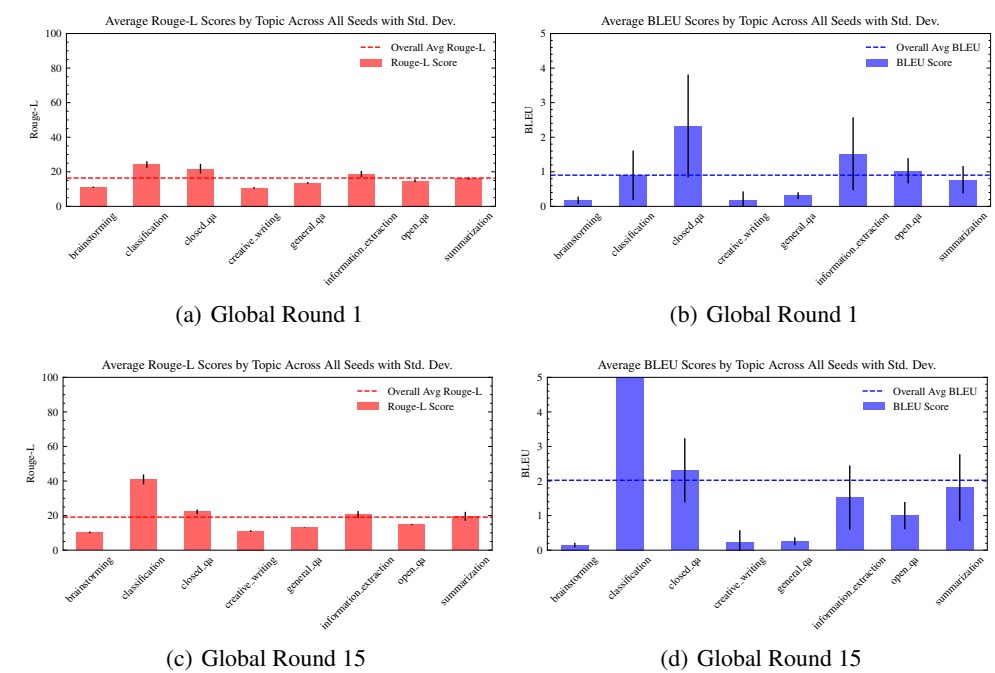

Figure 17: Rouge-L score distribution across different categories of Dolly dataset at global communication round 1 (a) and 15 (b) for FedSFT. Model:GPT-2

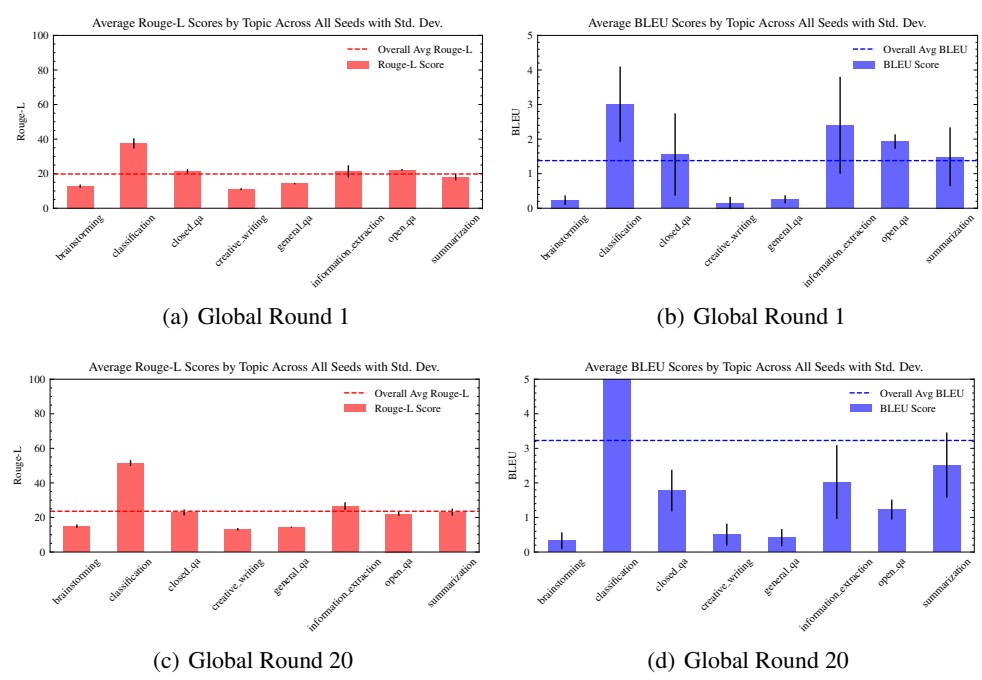

Figure 18: Rouge-L score distribution across different categories of Dolly dataset at global communication round 1 (a) and 20 (b) for FedSFT. Model: LLaMA.

may lack sufficient expertise or specialized knowledge needed to effectively generate or retrieve text that aligns with the high standards of content in these areas. By Global Round 15, while overall trends seem similar, several tasks show improved performance, narrowing the gap towards a higher

overall average score, denoted by the dashed line. Notably, tasks like "Markdown" and "Google Sheets" maintain high performance, and others like "Google Calendar" and "Google Meet" exhibit a noticeable improvement.

### D.3 Plots for Category Scores of S-NI

In Figure 21, 22, we plot the Rouge-L score and BLEU score from different categories on the S-NI dataset at round 1, 15 on GPT-2 model and round 1, 20 on LLaMA model. From the figures, we can find the tasks, including various natural language processing tasks, ranging from title generation to summarization and answer generation across different contexts. Notably, tasks that involve summarization (e.g., "task1540_parsed_pdfs_summarization", "task510_reddit_tifu_title_summarization") generally show lower performance, as seen by scores significantly below the overall average, represented by the dashed red line. In contrast, tasks focused on direct text generation show mixed results; some scores (e.g., "task1557_jfleg_answer_generation", "task402_grailqa_paraphrase_generation") are well above the average, indicating strong performance, while some other scores (e.g., "task1356_xlsum_title_generation", "task393_plausible_result_generation") fall below, suggesting areas needing improvement. Similar to the Dolly dataset, the error bars in rounds 15 and 20 appear generally smaller for most tasks, indicating a possible reduction in variability and increased model consistency over rounds.

## E Qualitative Study with Example Demonstration

We present the generations for each dataset as case studies. As shown in Tables 12, 13, 14, 15, 16, and 17, the generations of FedSFT is completely fluent and accurate from those of federated fine-tuning small small models.

Table 12: Generated responses from LLaMA family on Dolly dataset.

| Dolly | |
| --- | --- |
| Instruction | What is a dispersive prism? |
| Input | In optics, a dispersive prism is an optical prism that is used to disperse light, that is, to separate light into its spectral components (the colors of the rainbow). Different wavelengths (colors) of light will be deflected by the prism at different angles.[1] This is a result of the prism material's index of refraction varying with wavelength (dispersion). Generally, longer wavelengths (red) undergo a smaller deviation than shorter wavelengths (blue). The dispersion of white light into colors by a prism led Sir Isaac Newton to conclude that white light consisted of a mixture of different colors. |
| FedIT (7B) | In optics a dispersive prism used to separate light into its separate components wavelengths |
| FedIT+SFT (7B-13B) | In optics a dispersive prism separates light into its spectral components. |
| FedSFT (7B-13B) | In optics a dispersive prism are optical prisms that are used to disperse (or separate) light into the spectrum of the colors of the rainbow. |
| FedIT (13B) | A dispersive prism is a type of optical prism that defers light into its spectral components, that is, the colors in the rainbow. |
| Instruction | What is Magical Thinking? |
| Input | Magical thinking, or superstitious thinking,[1] is the belief that unrelated events are causally connected despite the absence of any plausible causal link between them, particularly as a result of supernatural effects.[1][2][3] Examples include the idea that personal thoughts can influence the external world without acting on them, or that objects must be causally connected if they resemble each other or have come into contact with each other in the past.[1][2][4] Magical thinking is a type of fallacious thinking and is a common sour |
| FedIT (7B) | Magical thinking is the belief that unconnected events are smartly connected. |
| FedIT+SFT (7B-13B) | As described, Magical Thinking is the believe that personal thoughts can influence external world without any causal actions performed |
| FedSFT (7B-13B) | Magical thinking would typically be referred to as "superstitious thinking." Example include, thinking that one's bedtime routine is exerting control over nightmares or thinking that a performance hat will make you a better golfer. |
| FedIT (13B) | Magical thinking is a cognitive behavior that imbues magical traits to objects or perceived social interactions. Magical thinking is an example of superstitious thinking[1] and is used to describe a superstition, while also containing non-related events (typically non-superstitious thinking) as causally connected despite the lack of a plausible causal connection between the events. |

## F  LLM USAGE

LLMs were employed for polishing the language. All technical contributions, derivations, and experiments were designed and validated by the authors. The authors take full responsibility for the accuracy and integrity of the content.

Table 13: Generated responses from GPT-2 family on Dolly dataset.

| | Dolly |
|---|---|
| Instruction | What is the average lifespan of a Golden Retriever? |
| Input | Golden Retrievers are a generally healthy breed; they have an average lifespan of 12 to 13 years.[9] Irresponsible breeding to meet high demand has led to the prevalence of inherited health problems in some breed lines, including allergic skin conditions, eye problems and sometimes snappiness.[9] These problems are rarely encountered in dogs bred from responsible breeders.[9] |
| FedIT (760M) | An average lifespan of 12 to 12 years for a Golden Retriever |
| FedIT+SFT (760M-1.5B) | According to breeders there average lifespan of a Golden is 13 years |
| FedSFT (760M-1.5B) | 12 to 13 years |
| FedIT (1.5B) | An average lifespan of 12 to 12.5 years. |
| FedOT (2-18-2) | Datation](http://www.treatyofPets.org/) provides a data source applied ... |
| Instruction | When was the 8088 processor released? |
| Input | The 8086[2] (also called iAPX 86)[3] is a 16-bit microprocessor chip designed by Intel between early 1976 and June 8, 1978, when it was released. The Intel 8088, released July 1, 1979,[4] is a slightly modified chip with an external 8-bit data bus (allowing the use of cheaper and fewer supporting ICs),[note 1] and is notable as the processor used in the original IBM PC design. |
| FedIT (760M) | The 8086 processor was released in December 1976. IBM released the 8086, an 16-bit microprocessor, in June 1978. The 8088, which was a modified chip, was released in July of 1979. |
| FedIT+SFT (760M-1.5B) | barring retro games, the 8088 was released July 1978. |
| FedSFT (760M-1.5B) | July 1, 1979 |
| FedIT (1.5B) | The 8088 processor was released in July 1979. |
| FedOT (2-18-2) | Alphabetically list the 8088 processors in order of release... |

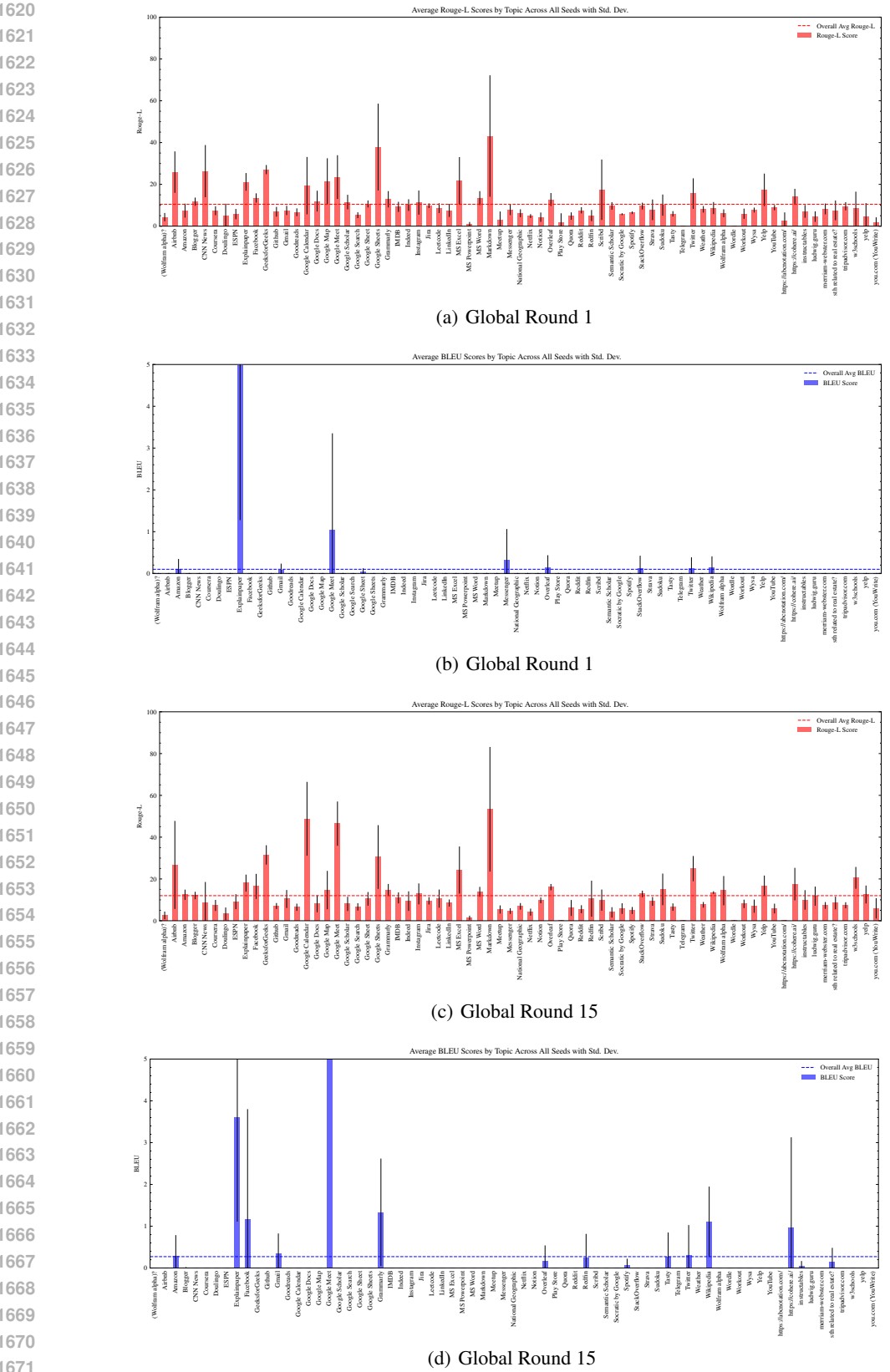

Figure 19: Rouge-L score distribution across different categories of SelfInst dataset at global communication round 1 (a) and 15 (b) for FedSFT. Model:GPT-2

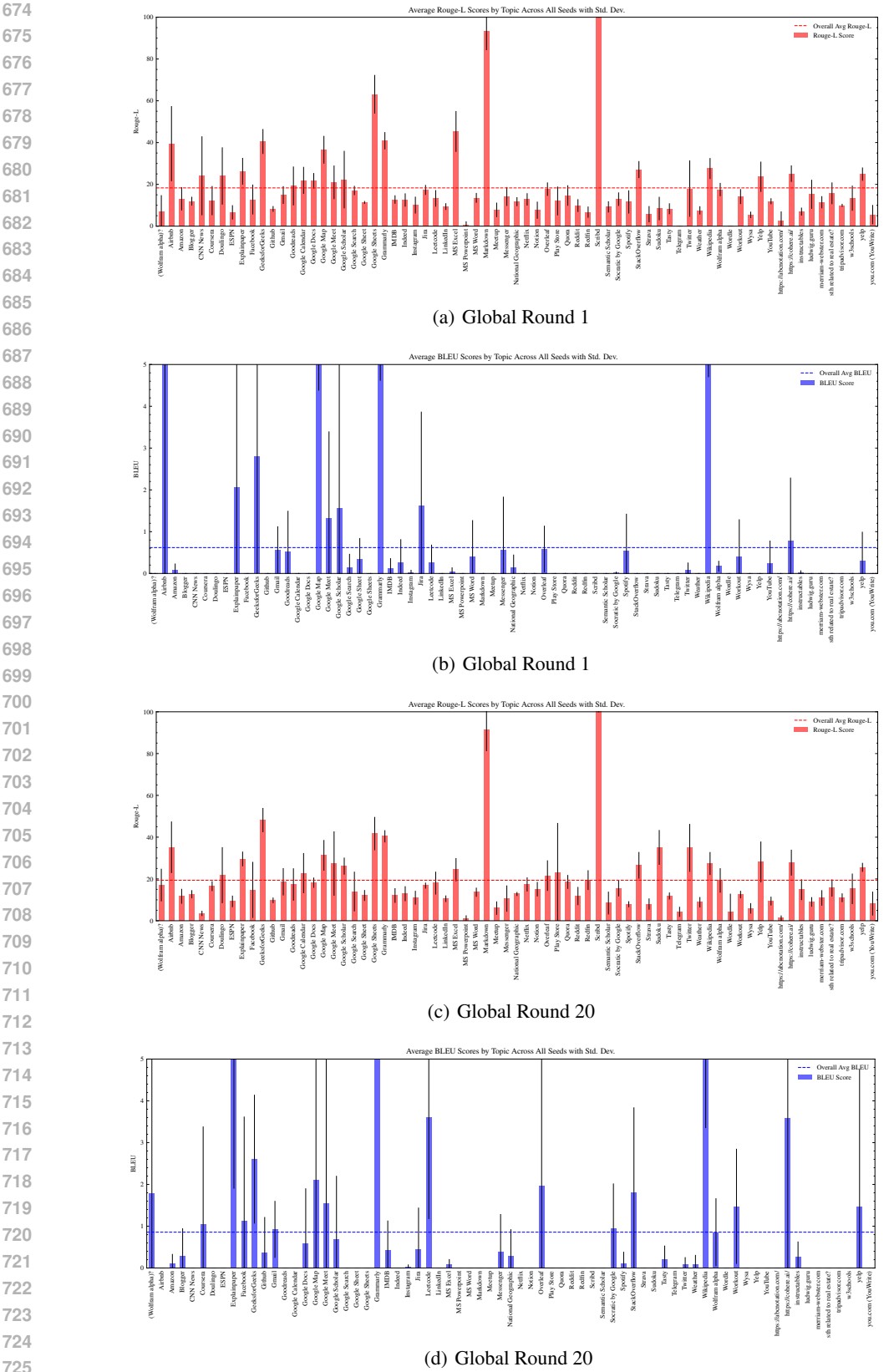

Figure 20: Rouge-L score distribution across different categories of SelfInst dataset at global communication round 1 (a) and 20 (b) for FedSFT. Model: LLaMA.

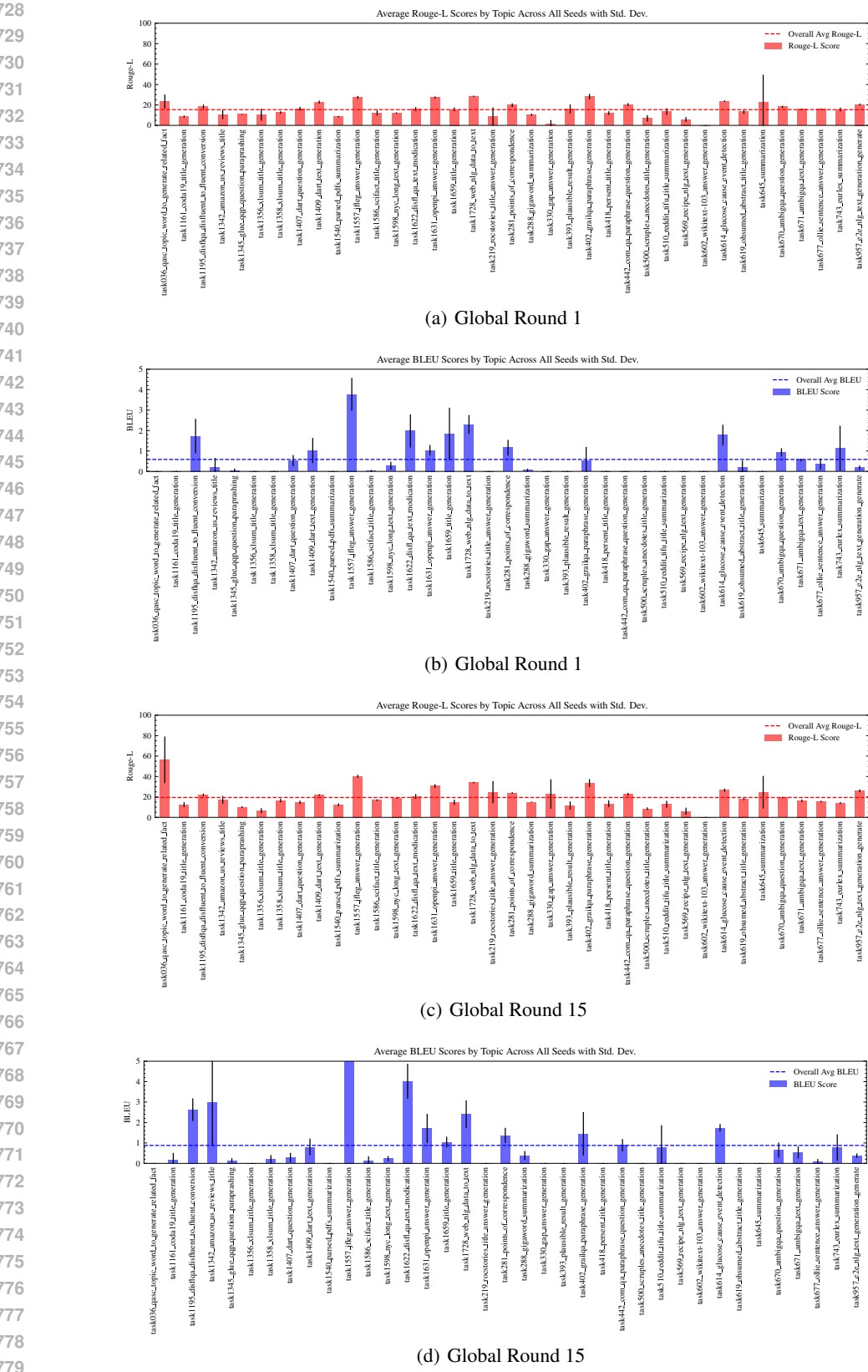

Figure 21: Rouge-L score distribution across different categories of S-NI dataset at global communication round 1 (a) and 15 (b) for FedSFT. Model: GPT-2

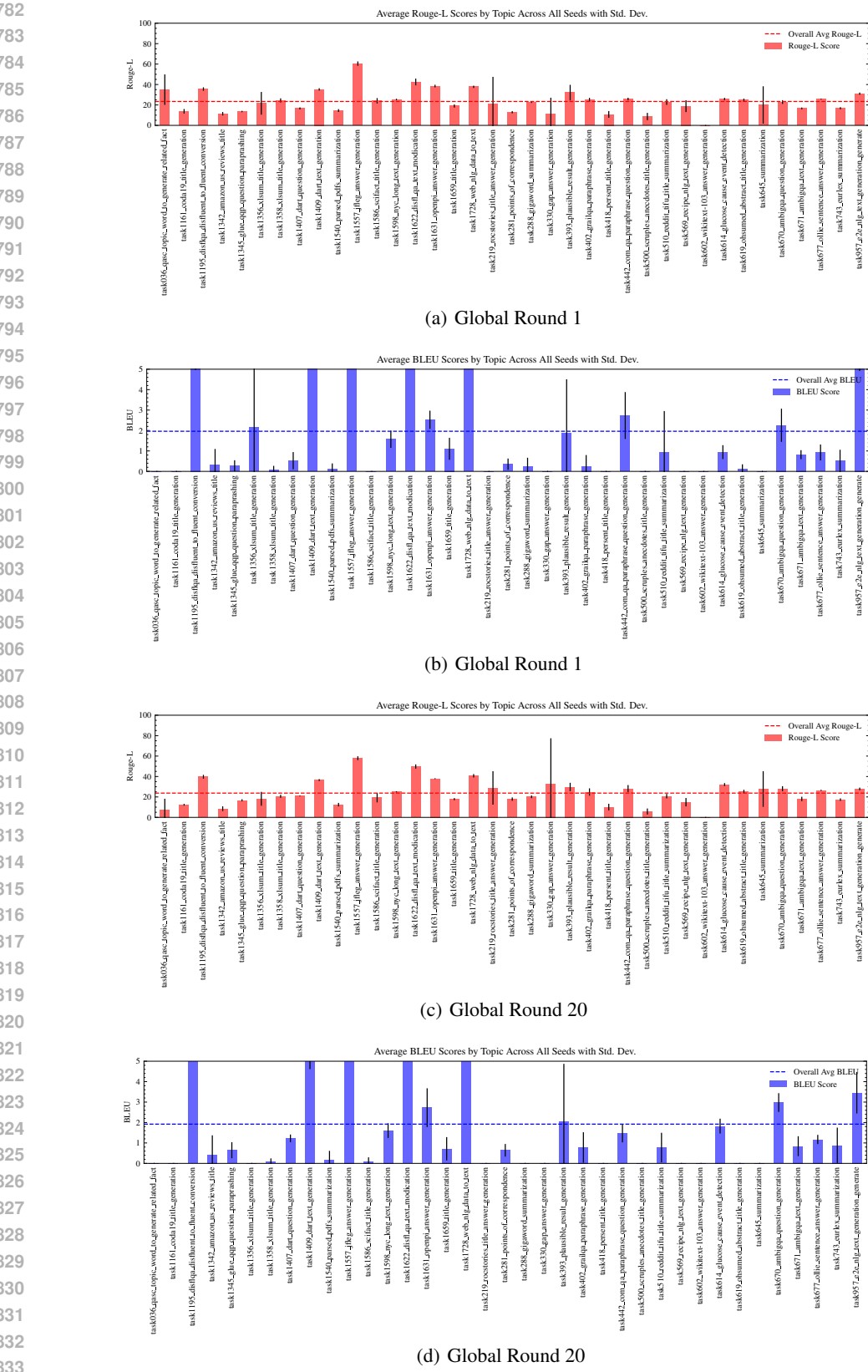

Figure 22: Rouge-L score distribution across different categories of S-NI dataset at global communication round 1 (a) and 20 (b) for FedSFT. Model: LLaMA.

Table 14: Generated responses from LLaMA family on SelfInst dataset.

| SelfInst | |
|---|---|
| Instruction | You will be required to write a few questions from the given company's interview section. |
| Input | Kohl's |
| FedIT (7B) | Which company do you think is having the best and worst interviews? Can you share the common interview questions in your company? Which field would you want your company to expand to to attract more candidates? |
| FedIT+SFT (7B-13B) | 1. What's your work culture like? 2. 3. What's your current expansion agenda in India? 4. What technology stack are you working on? 5. Do you have an ownership culture? |
| FedSFT (7B-13B) | 1. What made you interested in a career at Kohl's? 2. How did you hear about this role with Kohl's? 3. As you know, Kohl's is growing rapidly and there will be rigorous expectations for Extended Assisted Hours Supervisors and Associates to drive results - especially when we open new stores! Is this something you are prepared for? 4. Do you have any experience driving results? 5. What are some of the best practices you have around conflict management and team building? 6. What strengths can you bring to Kohl's in this role? 7. What values are important to you and how do you demonstrate these values in your everyday work? |
| FedIT (13B) | - Why apply for this job? - Tell me about a time when you focused on a goal and accomplished it. - What makes you a great team player? - Why us over other competitors? - How do you go above and beyond for people? - Why do you want to apply to this job? - Why Us? |
| Instruction | Identify the programming language used to write the given code. |
| Input | if (20 > 18)  printf("20 is greater than 18"); |
| FedIT (7B) | The programming language of the given code is Java |
| FedIT+SFT (7B-13B) | Python |
| FedSFT (7B-13B) | The language used appears to be the C language family, which supports conditional if statements without block bodies. |
| FedIT (13B) | The programming language is likely C, C++ or Java where comparing integers with the equal to or greater than symbol ('>') requires parenthesis. |

Table 15: Generated responses from GPT-2 family on SelfInst dataset.

| SelfInst | |
|---|---|
| Instruction | Assess the sentiment of the given tweet and categorize it as Positive, Negative, or Neutral. |
| Input | This is the best cheat-sheet paper that I've found in a LONG time. |
| FedIT (760M) | - Positive - Positive - Neutral - Negative - Neutral |
| FedIT+SFT (760M-1.5B) | Positive is Good, Negative is Not, Neutral is Bad |
| FedSFT (760M-1.5B) | Positive |
| FedIT (1.5B) | An expert cheat sheet for students, using numbers to explain overall reasoning |
| FedOT (2-18-2) | ASE stands for Automatic Teacher Interface, which does not require any human input or actions. |
| Instruction | Write the HTML syntax for the required action. |
| Input | if (20 > 18)  printf("20 is greater than 18"); |
| FedIT (760M) |  |
| FedIT+SFT (760M-1.5B) | noscript |
| FedSFT (760M-1.5B) |  |
| FedIT (1.5B) |  |
| FedOT (2-18-2) | Congrats!  You just once again coded an exception handler!  Now consider this code:... |

Table 16: Generated responses from LLaMA family on S-NI dataset.

| S-NI | |
|---|---|
| Instruction | In this task, we ask you convert a data table of restaurant descriptions into fluent natural-sounding English sentences. The input is a string of key-value pairs; the output should be a natural and grammatical English sentence containing all the information from the input. |
| Input | name[The Eagle], eatType[coffee shop], food[Italian], priceRange[cheap], customer rating[average], area[riverside], familyFriendly[no], near[Burger King] |
| FedIT (7B) | The Eagle is a family friendly Italian coffee shop located on a riverside in the historic core of Guilin. Its rich coffee ingredient and block art dessert are quite favored by locals and visitors. With a menu price range from affordable to moderate, the restaurant welcomes kids as long as they aren't too noisy. Popular food include sliced spicy Bos-Bosand block art art dessert which is a must-try. Overall, it is worthwhile because of their lovely shop and rich menu. |
| FedIT+SFT (7B-13B) | The Eagle is a coffeeshop and serves Italian food. The price range is cheap. The average family does not visit the restaurant. The restaurant is situated by the riverside, and is located near the Burger King. |
| FedSFT (7B-13B) | The Eagle is a coffee shop that serves Italian food at affordable prices in Riverside. The restaurant has only an average rating from its customers and it's not family friendly. Burger King is close to The Eagle. |
| FedIT (13B) | The Eagle is a locally popular coffee shop, popular among the folks in the neighborhood. It serves one of the best Italian food available at a fair price and is located on the riverside next to Burger King. |
| Instruction | In this task, you are given a part of an article. Your task is to generate headline (title) for this text. Preferred headlines are under fifteen words. |
| Input | (To appear in Theory and Practice of Logic Programming (TPLP)) ESmodels is designed and implemented as an experiment platform to investigate the semantics, language, related reasoning algorithms, and possible applications of epistemic specifications. We first give the epistemic specification language of ESmodels and its semantics. The language employs only one modal operator K but we prove that it is able to represent luxuriant modal operators by presenting transformation rules. Then, we describe basic algorithms and optimization approaches used in ESmodels. After that, we discuss possible applications of ESmodels in conformant planning and constraint satisfaction. Finally, we conclude with perspectives. |
| FedIT (7B) | ? Modal Epistemic Specification Inside Out ? Luxuriant Modal Logic OPERATIONS AND introduction of KLONE ? Algorithm and Optimization Approaches Conformant Planning and Constraint Satisfaction |
| FedIT+SFT (7B-13B) | - Exploring Epistemic Logic and Its Applications - Employing One But Powerful Modal Operator - Basic Algorithms Used in ESmodels - Possible Usages in Planning and Satisfiability - Concluding With Perspectives |
| FedSFT (7B-13B) | "ESmodels: An Experiment Platform Investigating Epistemic Specifications" |
| FedIT (13B) | 1. ESmodels is an experimental platform to investigate the semantics and language for epistemic specificaiton |

Table 17: Generated responses from GPT-2 family on S-NI dataset.

| S-NI | |
|---|---|
| Instruction | In this task, you will be given a short story. One sentence from the story is chosen. Consider the events that happen before that sentence, or are likely to have happened before it. Does any of them directly cause it, or simply make it possible? You should write your answer in the form "A >causes/enables> B". Try to use phrases and sentences from the story to compose your answer. |
| Input | story: John wanted to learn how to play the violin. He went to the music store to buy one. He selected a cheap model that he thought would do for a beginner. He started practicing and two strings broke right away. John went back to the store and bought a better violin. selected sentence: He started practicing and two strings broke right away. |
| FedIT (760M) | John was able to play violin because: a: The little violin he had was cheap. b: He liked playing the easy stuff. c: With two string breaks, he could play longer. d: He wanted to learn to play the violin. That was the reason he had gone to the store and bought one. |
| FedIT+SFT (760M-1.5B) | John wanted to learn how to start playing the violin. He went on to buy a cheap violin and started practicing. Two pieces of wood broke in front of him. He went for the second violin and it broke right away. |
| FedSFT (760M-1.5B) | John broke two strings because he bought a violin with a cheap beginner model. |
| FedIT (1.5B) | John seems to have experienced two different things before he learned how to play the violin, and during that time, two things happened that stopped him from practicing enough to rectify the issue. In order to display this, simply use phrases and sentences from John's story that pertain to what the story arewhases (or whatever term referring to what the story a twil pro to AND what happened to break the strings). For example, let's add a sentence such as, "John has experienced two events before he learned how to act." |
| FedOT (2-18-2) | Getting ready to write a response highest_of_three_strings:... |
| Instruction | In this task, you are given a text of the article. Your task is to generate a headline (title) for this article. |
| Input | the bank of japan appealed to financial markets to remain calm friday following the us decision to order daiwa bank ltd. to close its us operations . |
| FedIT (760M) | Bank had filed share resolution documents,Creditors backed bsma and several other regulators have measures in power to influence the bank's governance, Financial Markets said,Chongqing Prom finance director Wang Ma hoped that the Japan Bank Account Service Company, betraying no sign of confidence in banklandbanks people, instead Almaty today sought a voluntary settlement to resolve its internal problems The plan was only forced by a 25 minute hiccup of markets which blinded the whole world to a highly cautious bargain market outcome Not only is the market bid above mono, Banks in the current swath on Special explicit business plan economically insane and financial (wrote Louis Ivan Oct.IlB mouthpiece plunked in a billboard ).... |
| FedIT+SFT (760M-1.5B) | Bank of japan appeal to financial markets to remain calme friday following the me decision to order daiwaj bank ltd. to closures it's us operations |
| FedSFT (760M-1.5B) | Bank of japan appealed to the financial markets to remain calm. |
| FedIT (1.5B) | the bank of japan returned to financial markets to remain well positioned friday following the 'us decision to order daiwe bank ltd. to shut down' |
| FedOT (2-18-2) | Sir what are you doing pic.twitter.com/XyZQKQl9jf... |

