# OpenReview forum: "FedSFT: Resource-Constrained Federated Black-Box Adaptation of Large Language Models"
_ICLR.cc/2026/Conference — ICLR 2026 Conference Withdrawn Submission_

### Official Review · Reviewer_RoZa · 2025-10-26

**Soundness:** 2
**Presentation:** 2
**Contribution:** 2
**Rating:** 4
**Confidence:** 4

**Summary:**

The paper presents FedSFT, a federated surrogate fine-tuning framework designed to adapt black-box large language models in settings where clients have strict resource constraints and cannot access model parameters directly. Rather than fine-tuning the large model locally, each client fine-tunes a small surrogate model using LoRA, uploads only the low-rank deltas, and the server then aggregates these updates and constructs a composite model combining outputs of the untuned large model, untuned small model, and tuned small model. This composite model is used for server-side knowledge distillation, guiding future surrogate model updates. Experiments across GPT-2, OPT, and LLaMA architectures on sentiment control and instruction-following demonstrate that FedSFT maintains performance close to direct federated fine-tuning of large models while significantly reducing communication, computation, and memory overhead.

**Strengths:**

- Full-parameter federated fine-tuning of LLMs is infeasible for clients with only 4–8 GB VRAM, whereas FedSFT requires deploying only a lightweight surrogate model.
- FedSFT supports black-box settings where the full model weights are inaccessible, enabling fine-tuning of proprietary LLMs.
- The paper demonstrates strong empirical results across diverse experiments involving multiple model architectures and tasks.

**Weaknesses:**

- The server-side knowledge distillation relies on a public dataset that may come from a different domain, which could expose information about the task distribution or degrade performance in highly sensitive or domain-specific applications.
- The evaluation of non-IID data heterogeneity remains limited; although a Dirichlet split is included, the experiments do not fully capture more realistic cross-domain shifts or highly personalized client distributions.
- The method still requires access to the large model during inference, preventing clients from achieving standalone capability and reducing practicality in offline or low-connectivity environments.
- While the system is framed as privacy-preserving, there is no analysis of whether surrogate LoRA updates, logit offsets, or distillation signals could leak sensitive client information, nor are techniques such as secure aggregation or differential privacy considered to mitigate these risks.

**Questions:**

- The experiments are limited to only 10 clients, which may not adequately demonstrate scalability in realistic federated deployments. Testing with 100+ heterogeneous clients would better reflect practical system behavior and strengthen claims about efficiency and performance under scale.
- While the paper reports meaningful reductions in system costs (memory, communication, computation), there is no direct analysis of the accuracy–efficiency trade-off, making it difficult to fully assess how cost savings impact model capability.
- In Figure 5, as α increases, FedIT’s performance tends to decline more linearly than FedSFT; the paper should clarify why a larger α weakens baseline performance and provide theoretical or empirical justification for this behavior.
- In Figure 6, FedSFT generally outperforms alternatives as small-model size grows, but at 350M parameters FedIT appears stronger, which deserves explanation. Clarifying the underlying cause—such as insufficient surrogate capacity or instability in knowledge distillation when models are too small—would improve interpretability of the scaling results.

---

> ### Author Response · Authors · 2025-11-23
>
> We sincerely thank the reviewer for providing thoughtful and constructive feedback. To facilitate clear cross-referencing in our rebuttal, we denote weaknesses as **W1, W2**, …, questions as **Q1, Q2**, …, and our responses as **R1, R2**, ….
>
> **W1**: The server-side knowledge distillation relies on a public dataset that may come from a different domain, which could expose information about the task distribution or degrade performance in highly sensitive or domain-specific applications.
>
> **R1**: We want to clarify that the public dataset used for server-side distillation does not reveal any information about the private client tasks. Specifically, in our experiments, we use a small public dataset for distillation: 128 samples for GPT-2 and 512 samples for OPT/LLaMA that are randomly selected from the **Alpaca** dataset. It differs significantly from the client-side private **Dolly** dataset used for model fine-tuning. In practice, the public dataset used for distillation can be easily obtained using advanced generative AI techniques. For example, the **Alpaca** dataset is entirely generated by OpenAI's text-davinci-003 engine. Although domain-aligned data can improve distillation quality in classical settings, recent work on LLM distillation [1, 2] has shown that large, pre-trained models generalize well even when the distillation data is not domain-specific. This is because distillation for LLMs typically transfers general capabilities such as reasoning, instruction following, or response formatting, rather than domain-specific knowledge tied to private client data.
>
> Despite the limited size and domain mismatch of the public dataset, our experiments clearly demonstrate that our server-side distillation remains effective. This confirms that FedSFT does not rely on task-specific or privacy-sensitive data for distillation, and the use of a small synthetic dataset does not expose the underlying client task distribution.
>
> [1] Gu, Yuxian, et al. ``MiniLLM: Knowledge Distillation of Large Language Models." The Twelfth International Conference on Learning Representations.
>
> [2] Hsieh, Cheng-Yu, et al. ``Distilling step-by-step! outperforming larger language models with less training data and smaller model sizes." Findings of the Association for Computational Linguistics: ACL 2023. 2023.
>
> **W2**: The evaluation of non-IID data heterogeneity remains limited; although a Dirichlet split is included, the experiments do not fully capture more realistic cross-domain shifts or highly personalized client distributions.
>
> **R2**: This concern stems from overlooking the pathological non-IID experiment reported in the main paper, where each client is assigned data from only two exclusive categories. The Dirichlet distribution is widely used to simulate the non-IID data distribution in federated fine-tuning literature [3], and we follow the same setting to ensure a fair comparison with the baselines.
>
> Furthermore, we have already evaluated FedSFT under a pathological non-IID setting, where each client receives data from only two exclusive categories, as shown in Figure 10. This setup represents one of the most extreme forms of heterogeneity used in the FL literature and directly captures highly skewed and personalized client distributions. Our results in this setting further demonstrate the robustness of FedSFT under strong non-IID conditions. Due to the page limit, we present the results on pathological non-IID distribution in the main paper, and the results on the Dirichlet distribution in Appendix B.2.
>
> We have revised the relevant paragraph in Section 4.1 to reflect this clarification.
>
> [3] Zhang, Jianyi, et al. ``Towards building the federatedgpt: Federated instruction tuning." ICASSP 2024-2024 IEEE International Conference on Acoustics, Speech and Signal Processing (ICASSP). IEEE, 2024.
>
> **W3**: The method still requires access to the large model during inference, preventing clients from achieving standalone capability and reducing practicality in offline or low-connectivity environments.
>
> **R3**: This concern arises from a misunderstanding of our problem setting. The goal of our work is to enable the fine-tuning of a large model at the server when the training data are distributed across resource-constrained clients that cannot share their data due to privacy concerns. Therefore, instead of requiring clients to conduct standalone inference after fine-tuning, we want them to collaboratively adapt the large model while keeping all inferences on the server side. This setting aligns with the dominant deployment paradigm of modern LLM applications, where the large model is hosted on the server and clients interact with it through lightweight queries.

---

> ### Author Response · Authors · 2025-11-23
>
> **W4**: While the system is framed as privacy-preserving, there is no analysis of whether surrogate LoRA updates, logit offsets, or distillation signals could leak sensitive client information, nor are techniques such as secure aggregation or differential privacy considered to mitigate these risks.
>
> **R4**: We want to clarify that FedSFT does not introduce any additional privacy exposure beyond standard federated fine-tuning with LoRA methods. Clients only transmit the LoRA updates of the small model, and no logits or distillation signals are shared. The framework is privacy-preserving in the same sense as the standard FL: clients' raw training data remain entirely local and are never exposed to the server. Similar to standard FL frameworks, our method does not claim formal differential privacy guarantees, and thus we do not perform a theoretical privacy-leakage analysis in the current work.
>
> However, we want to emphasize that FedSFT is fully compatible with existing privacy-enhancing techniques, such as secure aggregation and differential privacy. These extensions can further mitigate potential leakage from shared LoRA updates and provide formal privacy guarantees if required.
>
> We have revised the paper to clarify the privacy scope of FedSFT and discussed how secure aggregation or differential privacy can be incorporated into our scheme to provide rigorous privacy guarantee in Section 3.1.
>
> **Q1**: The experiments are limited to only 10 clients, which may not adequately demonstrate scalability in realistic federated deployments. Testing with 100+ heterogeneous clients would better reflect practical system behavior and strengthen claims about efficiency and performance under scale.
>
> **R5**: To ensure a fair comparison, we follow the same experimental setting as FedIT [3], which evaluates all methods with 10 clients under full participation in a controlled cross-silo FL environment. This setting intentionally removes the randomness introduced by client sampling and allows for clear and reproducible comparisons of algorithmic behavior across methods. We agree that evaluating with a larger number of clients and partial participation would be valuable for studying cross-device FL scalability. However, as justified in [4], federated fine-tuning of LLMs is typically more prevalent in cross-silo FL settings. Therefore, we focus on the cross-silo setting, where the number of participating clients is limited and all clients participate in each round. We have revised Section 4.1 to clarify this design choice.
>
> [4] Zhuang, Weiming, Chen Chen, and Lingjuan Lyu. "When foundation model meets federated learning: Motivations, challenges, and future directions." arXiv preprint arXiv:2306.15546 (2023).
>
> **Q2**: While the paper reports meaningful reductions in system costs (memory, communication, computation), there is no direct analysis of the accuracy–efficiency trade-off, making it difficult to fully assess how cost savings impact model capability.
>
> **R6**: We agree that analyzing the accuracy-efficiency trade-off is important for fully understanding the practical benefits of FedSFT. To address this, we have provided a comprehensive comparison in Table 1, which jointly reports task performance and system cost under different client model sizes. The results show a consistent trend: larger client models achieve better performance but incur higher computation, communication, and VRAM overhead.
>
> **Table 1: Performance-efficiency trade-off under different client model sizes. We report ROUGE-L scores on Dolly, SelfInst, and S-NI. The server model is fixed at 13B.**
> | Method     | Size | Dolly | SelfInst | S-NI | Comm. Cost (MB) | Comp. Cost (TFLOPs) | VRAM (GB) |
> | ---------- | ---- | ----- | -------- | ---- | --------------- | ------------------- | --------- |
> | **FedSFT** | 350M | 20.7  | 12.0     | 21.7 | 1.4             | 0.4                 | 0.9       |
> |            | 1.3B | 21.8  | 15.2     | 25.8 | 3.0      | 1.4     | 3.0       |
> |            | 2.7B | 24.5  | 16.1     | 26.7 | 5.1      | 2.8       | 5.8       |
>
> We have added this result to Appendix C.9 in the revised manuscript.

---

> ### Author Response · Authors · 2025-11-23
>
> **Q3**: In Figure 5, as $\alpha$ increases, FedIT’s performance tends to decline more linearly than FedSFT; the paper should clarify why a larger $\alpha$ weakens baseline performance and provide theoretical or empirical justification for this behavior.
>
> **R7**: This is a misunderstanding of the figures as FedIT is a coefficient-independent baseline. In fact, FedIT does not use the composite model and therefore does not depend on $\alpha$; its performance should remain constant. It serves as the baseline for evaluating the effect of the composite model in FedSFT across different $\alpha$ values. To avoid confusion, we have revised Figure 5 by showing FedIT as a horizontal dashed line, clearly indicating that it is $\alpha$-independent. Only FedSFT varies with $\alpha$, which we clarify in the revised text and caption.
>
> **Q4**: In Figure 6, FedSFT generally outperforms alternatives as small-model size grows, but at 350M parameters FedIT appears stronger, which deserves explanation. Clarifying the underlying cause—such as insufficient surrogate capacity or instability in knowledge distillation when models are too small—would improve interpretability of the scaling results.
>
> **R8**: This is a misunderstanding of the figures as FedIT is a size-independent baseline. FedIT(13B) is plotted as a strong baseline that directly fine-tunes the large model and does not depend on the small model size. Therefore, its performance remains constant across different small model scales. To avoid confusion, we have revised Figure 6. So that FedIT(13B) is shown as a horizontal dashed line, clearly indicating that it is size-independent. Only FedIT+SFT and FedSFT rely on the small surrogate model capacity.

---

> > ### Comment · Reviewer_RoZa · 2025-11-23
> >
> > Thank you for providing the experiment and addressing some of my concerns, but I’ll keep my original point this time.

---

> > > ### Author Response · Authors · 2025-11-23
> > >
> > > Dear Reviewer,
> > >
> > > Thank you for acknowledging our rebuttal. If there are specific aspects of our response that you feel still fall short, we would greatly appreciate your feedback so that we can further clarify them. We believe that the additional experiments and explanations we have provided address the concerns you raised, and we kindly ask you to reconsider your evaluation if you find the clarifications helpful.

---

### Official Review · Reviewer_GGd7 · 2025-10-30

**Soundness:** 2
**Presentation:** 3
**Contribution:** 2
**Rating:** 4
**Confidence:** 4

**Summary:**

FedSFT is a black-box federated fine-tuning method where each client tunes a small LoRA-augmented surrogate; the server builds a composite teacher by adding the surrogate’s logit offsets to the frozen large model and distills this back into the surrogate for the next round, requiring only output token probabilities. Across LLMs and tasks, it matches direct federated fine-tuning while sharply reducing client costs (e.g., OPT per-round communication drops from 12.5 MB to 3 MB), making FL feasible on resource-constrained devices.

**Strengths:**

1. Much lower client costs (comm/computation/memory): e.g., per-round OPT communication drops from 12.5 MB → 3 MB, with overall reductions that make FL feasible on bandwidth- or memory-limited devices.
2. Extensive experiments have shown the effectiveness of the framework.

**Weaknesses:**

1. Sharing the logits of large models is a strong assumption. Also in the paper, it is mentioned that there will be a knowledge distillation on the server side. This is not realistic as well, this is like giving the model for free to the client.
2. Where does the performance gain come from. Is it because the composite/ensemble of several models? or the proposed training pipeline?

**Questions:**

see weaknesses.

---

> ### Author Response · Authors · 2025-11-23
>
> We sincerely thank the reviewer for providing thoughtful and constructive feedback.
>
> **Weakness 1**: Sharing the logits of large models is a strong assumption. Also in the paper, it is mentioned that there will be a knowledge distillation on the server side. This is not realistic as well, this is like giving the model for free to the client.
>
> **Response 1**: While our method assumes access to the full logits for clarity of presentation, the approach remains applicable when only partial logits are available. In practical deployments, commercial closed-source LLM APIs typically keep model parameters private but still return top-k logprobs. For example, the GPT-4 API exposes logprobs for the top five tokens [1]. To examine this setting, we conducted additional experiments in which the large model only returned top-5 logits. As shown in Table 1, FedSFT maintains strong performance under this constraint, demonstrating its applicability to real-world closed-source LLMs.
>
> **Table 1: Rouge-L scores (5 seeds) with top-5 logits from the large model for OPT models.
> A larger value indicates better performance.**
>
> | Method                         | Dolly       | SelfInst    | S-NI       |
> |--------------------------------|-------------|-------------|------------|
> | Base (13B)                     | 9.8 ± 0.2   | 6.7 ± 0.2   | 7.6 ± 0.1  |
> | FedIT (13B)                    | 23.6 ± 0.2  | 15.2 ± 0.7  | 25.9 ± 0.5 |
> | FedIT (1.3B)                   | 20.4 ± 0.5  | 11.5 ± 0.5  | 21.0 ± 0.1 |
> | FedIT + SFT (top-5) (1.3B–13B) | 23.4 ± 0.3  | 14.7 ± 0.4  | 30.7 ± 0.2 |
> | FedOT (2-2-2)                  | 6.1 ± 0.1   | 3.7 ± 0.1   | 3.8 ± 0.1  |
> | FedSFT (top-5) (1.3B–13B)      | 23.7 ± 0.3  | 15.3 ± 0.3  | 31.6 ± 0.3 |
>
> In the revised manuscript, we have added these new experimental results in Appendix C.7.
>
> Regarding the concern that server-side knowledge distillation is "like giving the model for free to the client,'' this concern arises from a misunderstanding of our workflow. In FedSFT, clients already possess the small-model architecture, and the server only returns updated small-model parameters after performing distillation locally. The entire distillation process occurs solely on the server. Clients never observe the large model’s parameters, logits, hidden states, or any intermediate computation. Their only interaction is uploading LoRA updates of their small models. Because no information from the large model is ever transmitted to clients, FedSFT does not leak the large model and does not provide clients with any signal that could be used to reconstruct or approximate it.
>
> [1] https://platform.openai.com/docs/api-reference/chat/create#chat-create-logprobs
>
> **Weakness 2**: Where does the performance gain come from. Is it because the composite/ensemble of several models? or the proposed training pipeline?
>
> **Response 2**: We have already quantified the contribution of each core component in Table 1 of the main paper. To make this clearer, we now highlight these results explicitly. The performance gain in FedSFT comes from two core components: 1) the composite model; and 2) the knowledge distillation based on the composite model.
>
> To isolate the contribution of the composite model, we have included a baseline named FedIT+SFT, which follows the same training process as FedIT (standard FL paradigm under the same client resource limitations) but uses the composite model for inference. By comparing FedIT+SFT with FedIT, we observe an average relative improvement of 7.4\% across all models and datasets in Table 2, clearly demonstrating the effectiveness of the composite model component.
>
> To isolate the contribution of knowledge distillation, we compare FedSFT with FedIT+SFT, where the only difference is that FedSFT additionally performs knowledge distillation in Table 2. This comparison directly quantifies the benefit of the distillation stage, showing an average relative improvement of 4.6\% across all models and datasets.
>
> **Table 2: Rouge-L scores (5 seeds). A larger value indicates better performance.**
> | Model     | Method     | Dolly          | SelfInst       | S-NI    |
> | --------- | ---------------------- | -------------- | -------------- | -------------- |
> | **OPT**   | FedIT (1.3B)           | 20.4 ± 0.5     | 11.5 ± 0.5    | 21.0 ± 0.1    |
> |     | FedIT+SFT (1.3B–13B)   | 21.5 ± 0.3     | 13.0 ± 0.4     | 23.3 ± 0.2    |
> |       | **FedSFT (1.3B–13B)**  | **21.8 ± 0.2** | **15.1 ± 0.6** | **25.8 ± 0.2** |
> | **GPT-2** | FedIT (760M)           | 17.8 ± 0.5     | 10.4 ± 0.3     | 18.4 ± 0.3    |
> |    | FedIT+SFT (760M–1.5B)  | 18.6 ± 0.4     | 10.9 ± 0.5     | 21.4 ± 0.3    |
> |     | **FedSFT (760M–1.5B)** | **18.9 ± 0.5** | **11.0 ± 0.4** | **21.6 ± 0.2** |
> | **LLaMA** | FedIT (7B)             | 23.3 ± 0.6     | 17.6 ± 0.6     | 25.9 ± 0.2    |
> |     | FedIT+SFT (7B–13B)     | 23.5 ± 0.7     | 18.9 ± 0.3     | 26.7 ± 0.4    |
> |     | **FedSFT (7B–13B)**    | **23.8 ± 0.4** | **19.1 ± 0.7** | **28.7 ± 0.3** |

---

### Official Review · Reviewer_oRdb · 2025-11-03

**Soundness:** 3
**Presentation:** 3
**Contribution:** 2
**Rating:** 4
**Confidence:** 3

**Summary:**

The paper presents FedSFT, a framework for federated adaptation of large language models in cases where model parameters cannot be accessed. Each client trains a small local model using LoRA, and the server aggregates these updates to build a composite model that combines the black-box LLM with the aggregated surrogates through knowledge distillation. The aim is to make collaborative model tuning possible under limited resources and privacy constraints.

**Strengths:**

•  The paper tackles a timely and relevant problem: adapting large models in federated settings when direct access to parameters is restricted.
•  The overall design—client-side LoRA fine-tuning combined with server-side distillation—is simple and well motivated, and achieves strong efficiency gains while maintaining good accuracy.
•  The motivation and formulation are clearly articulated, and the framework could inspire future work on privacy-aware model adaptation.
•  Experimental results show substantial reductions in resource usage (4–9×) without large performance drops, highlighting potential practical value.

**Weaknesses:**

•  The “black-box” assumption conflicts with the claim that the model can provide full output logits, which real API-based systems (e.g., GPT-4, Claude) do not expose.
•  The experimental comparisons are not capacity-matched: FedSFT fine-tunes a 1.3B surrogate model, while baselines fine-tune 13B models. This mismatch weakens the claim of “comparable performance with higher efficiency.”
•  Important baselines are missing—such as small-model FedLoRA without distillation or a centralized distillation variant—making it unclear which part of the framework drives the improvements.
•  Overall, the work reads as an engineering composition of existing ideas (federated learning, LoRA, and distillation) rather than a significant methodological advance.

**Questions:**

•  Black-box assumption realism: Section 3.1 assumes that the black-box LLM can return full logits. In practical API settings this is unrealistic—would the method still function with only sampled text outputs?
•  Baseline fairness and model capacity: All reported baselines use a 13B model, whereas FedSFT uses a 1.3B surrogate. Are there results using the same model size for a fairer comparison?
•  Ablation and component contribution: The current ablations vary α and model scale but do not isolate core components. Since LoRA and aggregation mainly support the distillation stage, can the authors clarify how much each part contributes to overall performance?
•  Centralized vs federated benefits: Given that the experiments are simulated on a single machine, what is the actual gain from federated aggregation compared with a centralized distillation setting?

---

> ### Author Response · Authors · 2025-11-23
>
> We sincerely thank the reviewer for providing thoughtful and constructive feedback. To facilitate clear cross-referencing in our rebuttal, we denote weaknesses as **W1, W2**, …, questions as **Q1, Q2**, …, and our responses as **R1, R2**, ….
>
> **W1**: The “black-box” assumption conflicts with the claim that the model can provide full output logits, which real API-based systems (e.g., GPT-4, Claude) do not expose.
>
> **R1**: Our use of the term “black-box” follows the standard definition in machine learning: a model is considered black-box when its internal information, such as its parameters, gradients, and architecture, is inaccessible, while its outputs can still be queried. This is consistent with prior works [1,2], which also claim large language models as black-box models while relying on their output logits. In practical deployments, commercial closed-source LLM APIs typically keep model parameters private but still return top-k logprobs. For example, the GPT-4 API exposes logprobs for the top five tokens [3]. While our method assumes access to the full logits for clarity of presentation, the approach remains applicable when only partial logits are available. To validate this, we conducted additional experiments in which the large model only returned top-5 logits. As shown in Table 1, FedSFT maintains strong performance under this constraint, demonstrating its applicability to real-world closed-source LLMs.
>
> **Table 1: Rouge-L scores (5 seeds) with top-5 logits from the large model for OPT models.
> A larger value indicates better performance.**
>
> | Method                         | Dolly       | SelfInst    | S-NI       |
> |--------------------------------|-------------|-------------|------------|
> | Base (13B)                     | 9.8 ± 0.2   | 6.7 ± 0.2   | 7.6 ± 0.1  |
> | FedIT (13B)                    | 23.6 ± 0.2  | 15.2 ± 0.7  | 25.9 ± 0.5 |
> | FedIT (1.3B)                   | 20.4 ± 0.5  | 11.5 ± 0.5  | 21.0 ± 0.1 |
> | FedIT + SFT (top-5) (1.3B–13B) | 23.4 ± 0.3  | 14.7 ± 0.4  | 30.7 ± 0.2 |
> | FedOT (2-2-2)                  | 6.1 ± 0.1   | 3.7 ± 0.1   | 3.8 ± 0.1  |
> | FedSFT (top-5) (1.3B–13B)      | 23.7 ± 0.3  | 15.3 ± 0.3  | 31.6 ± 0.3 |
>
> In the revised manuscript, we have added these new experimental results in Appendix C.7.
>
> [1] Shi, Weijia, et al. "Detecting pretraining data from large language models." 12th International Conference on Learning Representations, ICLR 2024.
>
> [2] Carlini, Nicholas, et al. "Stealing part of a production language model." International Conference on Machine Learning. PMLR, 2024.
>
> [3] https://platform.openai.com/docs/api-reference/chat/create#chat-create-logprobs
>
>
> **W2**: The experimental comparisons are not capacity-matched: FedSFT fine-tunes a 1.3B surrogate model, while baselines fine-tune 13B models. This mismatch weakens the claim of “comparable performance with higher efficiency.”
>
> **R2**: This concern arises from a misunderstanding of our setting. The goal of our work is to enable the fine-tuning of a target 13B model at the server when the training data are distributed across resource-constrained clients that cannot share their data due to privacy concerns.
>
> In the traditional FL paradigm, each client must download and fine-tune the full 13B model locally before sending updates to the server. However, this setup is infeasible in our scenario: clients are unable to store or train such a large model due to strict memory and computational limits, and in many practical settings, the target model’s parameters (e.g., for a proprietary 13B model) may not be accessible to the clients.
>
> FedSFT is specifically designed to address these constraints. Each client fine-tunes a small surrogate model (1.3B) on its local data and uploads the resulting model to the server. The server then aggregates these small models and leverages them to steer the target 13B model, effectively transferring the knowledge learned from the clients to the large model without requiring clients to access or train it directly.
>
> In our experiments, the comparisons between our method and the baseline are made on the same 13B target model after tuning. Our results show that FedSFT achieves comparable performance to methods that fine-tune the full 13B model locally, while requiring orders of magnitude less local computation and communication. This demonstrates that FedSFT achieves the claimed goal of "comparable performance with much higher efficiency.''

---

> ### Author Response · Authors · 2025-11-23
>
> **W3**: Important baselines are missing-such as small-model FedLoRA without distillation or a centralized distillation variant—making it unclear which part of the framework drives the improvements.
>
> **R3**: We would like to clarify that the *small-model FedLoRA without any distillation baseline* is already included in our experiments, implemented as FedIT(1.3B), FedIT(760M), and FedIT(7B) for OPT, GPT-2, and LLaMA, respectively.
>
> Regarding the "centralized distillation'' baseline, such a setting would assume that all client data can be centralized at the server, which directly violates the data privacy and decentralization assumptions fundamental to the FL paradigm. Nonetheless, we acknowledge that such a baseline could serve as an upper-bound reference for our method's performance when data privacy constraints are relaxed. To further strengthen our study, we have added a centralized baseline, denoted as CentralSFT, which performs centralized fine-tuning on the small model followed by distillation using a composite model on the server side.
>
> In addition to CentralSFT, as explained later in response to **Q3**, we also include a local-only baseline, denoted as LocalSFT, where each client fine-tunes its small model independently, followed by distillation using a composite model on the server side. Together, CentralSFT and LocalSFT further strengthen our existing ablation study and isolate the contribution of model aggregation in FedSFT.
>
> As shown in Table 2, FedSFT consistently outperforms the local-only baseline (LocalSFT) and approaches the centralized baseline (CentralSFT) across all considered datasets.
>
> **Table 2: Rouge-L scores (5 seeds) for OPT models. A larger value indicates better performance.**
>  | Method                | Dolly          | SelfInst       | S-NI           |
>  | --------------------- | -------------- | -------------- | -------------- |
>  | LocalSFT (1.3B–13B)   | 20.7 ± 1.1     | 11.9 ± 0.8     | 20.9 ± 2.1     |
>  | CentralSFT (1.3B–13B) | 23.7 ± 0.3     | 15.5 ± 0.8     | 26.1 ± 0.2     |
>  | **FedSFT (1.3B–13B)** | **21.8 ± 0.2** | **15.1 ± 0.6** | **25.8 ± 0.2** |
>
> We have added the experimental results of both the centralized baseline and local baseline in Appendix C.8.
>
> **W4**: Overall, the work reads as an engineering composition of existing ideas (federated learning, LoRA, and distillation) rather than a significant methodological advance.
>
> **R4**: We respectfully disagree with the statement that our work is merely an engineering composition of existing ideas. Although our framework leverages elements such as federated learning, LoRA, and distillation, the core contribution is not a direct combination of these techniques. Instead, we introduce a **new federated surrogate fine-tuning framework** tailored to the unique and previously unaddressed setting of fine-tuning a black-box large model. This problem setup is fundamentally different from conventional FL or standard distillation, and existing methods cannot be directly applied.
>
> Specifically, our technical contributions include:
> 1) **A principled composite-model formulation grounded in an RL-based fine-tuning objective.** In Eq. (5), we cast the fine-tuning of a large black-box model as a reinforcement learning problem that maximizes task reward while constraining divergence from the untuned reference model. Building on this formulation, Theorem 1 shows that the next-token distribution of the fine-tuned model can be represented by a composite model whose logits combine contributions from the untuned large model, the small untuned model, and the small fine-tuned model. This composite-model construction is new and does not follow from existing FL or distillation techniques.
>
> 2) **A novel distillation mechanism enabled by the composite model.** Unlike standard distillation—in which the teacher is an explicitly fine-tuned large model—our teacher is implicitly defined via the composite model that approximates the behavior of the (inaccessible) fine-tuned large LM. This allows knowledge transfer to small models without requiring parameter access or fine-tuning of the large model, which is a key technical innovation enabling federated fine-tuning under the black-box constraint.
>
> Overall, our framework introduces both a new problem setting and new technical machinery to enable fine-tuning of closed-source LLMs in federated environments, going beyond a straightforward combination of existing ideas.
>
> **Q1**: Black-box assumption realism: Section 3.1 assumes that the black-box LLM can return full logits. In practical API settings this is unrealistic—would the method still function with only sampled text outputs?
>
> **R5**: As discussed in **R1**, FedSFT still functions even when the black-box large model returns only the top few logits, as is the case with current closed-source LLMs (e.g., GPT-4). Please see R1 for details.

---

> ### Author Response · Authors · 2025-11-23
>
> **Q2**: Baseline fairness and model capacity: All reported baselines use a 13B model, whereas FedSFT uses a 1.3B surrogate. Are there results using the same model size for a fairer comparison?
>
> **R6**: We believe this concern arises from a misunderstanding. As discussed in **R2**, we indeed use the same model size for a fair comparison.
>
> **Q3**: Ablation and component contribution: The current ablations vary $\alpha$ and model scale but do not isolate core components. Since LoRA and aggregation mainly support the distillation stage, can the authors clarify how much each part contributes to overall performance?
>
> **R7**: LoRA fine-tuning has been used in both our approach and all the other baselines in the experiments for fair comparison. It only serves as an exemplary fine-tuning strategy rather than a unique component of FedSFT. Therefore, ablating LoRA will not provide meaningful insight into the contribution of our framework, and any other PEFT methods could be used here to replace LoRA.
>
> Regarding aggregation, we have added a local-only baseline, denoted as LocalSFT(1.3B-13B). In this setting, each client fine-tunes its own 1.3B model independently. The server then performs knowledge distillation using the composite of pre-trained 13B model and fine-tuned 1.3B model to produce an updated 1.3B model, which is sent back to clients for the next round of training. As shown in Table 2, FedSFT consistently outperforms the local-only baseline (LocalSFT) and approaches the centralized baseline (CentralSFT) across all datasets. Further details of this ablation study can be found in R3.
>
> We have already quantified the contribution of each core component in Table 1 of the main paper. To make this clearer, we now highlight these results explicitly. Our method has two core components: 1) the composite model and 2) the knowledge distillation based on the composite model.
>
> To isolate the contribution of the composite model, we have included a baseline named FedIT+SFT, which follows the same training process as FedIT but uses the composite model for inference. By comparing FedIT+SFT with FedIT, we observe an average relative improvement of 7.4\% across all models and datasets in Table 3, clearly demonstrating the effectiveness of the composite model component.
>
> To isolate the contribution of knowledge distillation, we compare FedSFT with FedIT+SFT, where the only difference is that FedSFT additionally performs knowledge distillation. This comparison directly quantifies the benefit of the distillation stage, showing an average relative improvement of 4.6\% across all models and datasets in Table 3.
>
> **Table 3: Rouge-L scores (5 seeds). A larger value indicates better performance.**
> | Model | Method| Dolly | SelfInst | S-NI |
> | --------- | ---------------------- | -------------- | -------------- | -------------- |
> | **OPT**   | FedIT (1.3B)           | 20.4 ± 0.5     | 11.5 ± 0.5     | 21.0 ± 0.1  |
> |    | FedIT+SFT (1.3B–13B)   | 21.5 ± 0.3     | 13.0 ± 0.4     | 23.3 ± 0.2  |
> |    | **FedSFT (1.3B–13B)**  | **21.8 ± 0.2** | **15.1 ± 0.6** | **25.8 ± 0.2** |
> | **GPT-2** | FedIT (760M)  | 17.8 ± 0.5     | 10.4 ± 0.3     | 18.4 ± 0.3  |
> |     | FedIT+SFT (760M–1.5B)  | 18.6 ± 0.4     | 10.9 ± 0.5     | 21.4 ± 0.3  |
> |    | **FedSFT (760M–1.5B)** | **18.9 ± 0.5** | **11.0 ± 0.4** | **21.6 ± 0.2** |
> | **LLaMA** | FedIT (7B)  | 23.3 ± 0.6 | 17.6 ± 0.6   | 25.9 ± 0.2  |
> |   | FedIT+SFT (7B–13B)   | 23.5 ± 0.7 | 18.9 ± 0.3  | 26.7 ± 0.4  |
> |   | **FedSFT (7B–13B)**    | **23.8 ± 0.4** | **19.1 ± 0.7** | **28.7 ± 0.3** |
>
> **Q4**: Centralized vs federated benefits: Given that the experiments are simulated on a single machine, what is the actual gain from federated aggregation compared with a centralized distillation setting?
>
> **R8**: To further strengthen our study, we have added a centralized baseline, denoted as CentralSFT, which performs centralized fine-tuning on the small model, followed by knowledge distillation using the composite model on the server. This baseline assumes that all client data can be pooled on the server, an assumption that directly violates the data privacy and decentralization principles fundamental to the FL paradigm. Consequently, this centralized setting serves as an upper bound for our method.
>
> The experimental results are shown in Table 2. We can observe that FedSFT approaches the upper-bound performance achieved by CentralSFT in the centralized setting while preserving data privacy. More detailed discussions can be found in **R3**.
>
> **Table 2: Rouge-L scores (5 seeds) for OPT models. A larger value indicates better performance.**
>  | Method                | Dolly          | SelfInst       | S-NI           |
>  | --------------------- | -------------- | -------------- | -------------- |
>  | LocalSFT (1.3B–13B)   | 20.7 ± 1.1| 11.9 ± 0.8 | 20.9 ± 2.1 |
>  | CentralSFT (1.3B–13B) | 23.7 ± 0.3| 15.5 ± 0.8 | 26.1 ± 0.2 |
>  | **FedSFT (1.3B–13B)** | **21.8 ± 0.2** | **15.1 ± 0.6** | **25.8 ± 0.2** |

---

### Author Response · Authors · 2025-11-30
**Global Response for Area Chairs**

(11) Concern on accuracy–efficiency trade-off (R3: Q2).

We have provided a comprehensive comparison in **Table 4**, which jointly reports task performance and system cost under different client model sizes. The results show a consistent trend: larger client models achieve better performance but incur higher computation, communication, and VRAM overhead.

**Table 4: Performance-efficiency trade-off under different client model sizes. We report ROUGE-L scores on Dolly, SelfInst, and S-NI. The server model is fixed at 13B.**
| Method     | Size | Dolly | SelfInst | S-NI | Comm. Cost (MB) | Comp. Cost (TFLOPs) | VRAM (GB) |
| ---------- | ---- | ----- | -------- | ---- | --------------- | ------------------- | --------- |
| **FedSFT** | 350M | 20.7  | 12.0     | 21.7 | 1.4    | 0.4     | 0.9   |
|    | 1.3B | 21.8  | 15.2     | 25.8 | 3.0             | 1.4        | 3.0       |
|    | 2.7B | 24.5  | 16.1     | 26.7 | 5.1             | 2.8       | 5.8       |

We have added this result to Appendix C.9 in the revised manuscript.

(12) Concern on Figures 5 and 6 (R3: Q3, Q4).

This is a misunderstanding of the figures as FedIT is a size- and coefficient-independent baseline. In both figures, we revised the plots to show FedIT (or FedIT-13B) as a horizontal dashed line and updated the captions accordingly in Section 4.5, making it explicit that only FedIT+SFT and FedSFT vary with surrogate capacity or the composite-model coefficient.

**Summary of Revisions**

In summary, the reviewers’ concerns about the black-box assumption stem from terminology differences: in our setting, a model is considered black-box when its parameters are inaccessible, even if its logits are available. We also added new top-5 logit experiments showing that FedSFT remains effective under the limited-access setting for state-of-the-art closed-source models such as GPT-4.

Other concerns do not affect our main claims. We clarified baseline fairness, added centralized and local baselines, highlighted the component isolation analysis already in the main paper, emphasized both Dirichlet and pathological non-IID settings, and explained that FedSFT introduces no additional privacy exposure and aligns with the standard cross-silo FL setup.

---

### Author Response · Authors · 2025-11-30
**Global Response for Area Chairs**

(5) Concern on isolating the contributions of core components (R1: W4, Q3; R2: W2).

The reviewers missed the analysis isolating the two components of FedSFT, which is already reported in Table 1 of the main paper. We now highlight this result more explicitly in the revision.

FedSFT consists of two core components: 1) the composite model and 2) the composite-model-based distillation. To isolate the composite model's contribution, we introduced the FedIT+SFT baseline, which applies the composite model without distillation; this yields a 7.4\% average improvement over FedIT, as shown in **Table 3**. To isolate distillation, we compare FedSFT with FedIT+SFT, where the only difference is the added distillation step, leading to an additional 4.6\% average improvement in **Table 3**.

**Table 3: Rouge-L scores (5 seeds). A larger value indicates better performance.**
|Model|Method|Dolly|SelfInst|S-NI|
|--|--|-|-|-|
|**OPT**|FedIT(1.3B)|20.4±0.5|11.5±0.5|21.0±0.1|
| |FedIT+SFT(1.3B–13B)|21.5±0.3|13.0±0.4|23.3±0.2|
| |**FedSFT(1.3B–13B)**|**21.8±0.2**|**15.1±0.6**|**25.8±0.2**|
| **GPT-2** |FedIT(760M)|17.8±0.5|10.4± 0.3|18.4±0.3|
| |FedIT+SFT(760M–1.5B)|18.6±0.4|10.9±0.5|21.4±0.3|
| |**FedSFT(760M–1.5B)**|**18.9±0.5**|**11.0±0.4**|**21.6±0.2**|
|**LLaMA**|FedIT(7B)|23.3±0.6|17.6±0.6|25.9±0.2|
| |FedIT+SFT(7B–13B)|23.5±0.7|18.9±0.3|26.7±0.4|
| |**FedSFT (7B–13B)**|**23.8±0.4**|**19.1±0.7**|**28.7±0.3**|

(6) Concern on distillation data and task-distribution exposure (R3: W1).

We clarified that the small public dataset used for server-side distillation (128 samples for GPT-2, 512 for OPT/LLaMA) is entirely synthetic and domain-mismatched from client data. We added explanations and citations showing that LLM distillation transfers general reasoning and instruction-following ability rather than domain-specific knowledge. Although domain-aligned data can improve distillation quality, recent work on LLM distillation [1, 2] has shown that large, pre-trained models generalize well even when the distillation data is not domain-specific. This is because distillation for LLMs typically transfers general capabilities such as reasoning, instruction following, or response formatting, rather than domain-specific knowledge tied to private client data.

[1] Gu, Yuxian, et al. ``MiniLLM: Knowledge Distillation of Large Language Models." The Twelfth International Conference on Learning Representations.

[2] Hsieh, Cheng-Yu, et al. ``Distilling step-by-step! outperforming larger language models with less training data and smaller model sizes." Findings of the Association for Computational Linguistics: ACL 2023.

(7) Concern on non-IID data heterogeneity evaluation (R3: W2).

The reviewer missed the pathological non-IID experiment already reported in the main paper, where each client receives only two exclusive categories. We clarified that we follow the standard Dirichlet-based non-IID setting used in prior federated fine-tuning work [3]. In addition, we have already evaluated FedSFT under a severe pathological non-IID setting (each client receives only two exclusive categories), capturing extreme personalization.

We revised Section 4.1 to emphasize both settings.

[3] Zhang, Jianyi, et al. ``Towards building the federatedgpt: Federated instruction tuning." IEEE ICASSP 2024.

(8) Concern on requiring access to the large model during inference (R3: W3).

We clarified that FedSFT is designed for the realistic deployment setting where inference is executed at the server, and clients interact with the large model only through lightweight queries. Our goal is not to enable standalone client-side inference but to collaboratively fine-tune a server-hosted LLM without exposing client data.

(9) Concern on privacy exposure and lack of formal guarantees (R3: W4).

We emphasize that our work focuses on improving the resource efficiency and performance of federated fine-tuning of LLMs on resource-constrained edge devices, rather than providing rigorous privacy guarantees, which represents a separate line of research. To answer the reviewers' questions, we clarified that FedSFT introduces no additional privacy exposure beyond standard FL with LoRA updates. We also added an explanation in Section 3.1 that FedSFT is fully compatible with secure aggregation and differential privacy, which can be integrated to provide formal guarantees if desired.

(10) Concern on the limited number of clients (R3: Q1).

We followed the same experimental setting as FedIT [3] and other closely related works to enable fair comparisons under the cross-silo FL settings with 10 clients. While large-scale evaluations are important for cross-device FL, prior work [4] shows that federated LLM fine-tuning is typically a cross-silo scenario. We clarified the cross-silo setting in Section 4.1.

[4] Zhuang, Weiming, Chen Chen, and Lingjuan Lyu. "When foundation model meets federated learning: Motivations, challenges, and future directions." arXiv preprint arXiv:2306.15546.

---

### Author Response · Authors · 2025-11-30
**Global Response for Area Chairs**

We sincerely thank the AC, SAC, PC, and all reviewers for their constructive comments and valuable feedback.

Our paper addresses a key practical challenge in federated fine-tuning of LLMs under real-world constraints, where 1) **parameter access is unavailable** in proprietary LLMs, and 2) clients lack sufficient **computational, communication, and memory** resources to fine-tune LLMs locally.

To tackle these limitations, we proposed **FedSFT**, a **novel federated black-box fine-tuning of LLMs framework** that requires access only to the logits of output sequences and significantly reduces resource demands on clients. In each communication round, clients only fine-tune a lightweight surrogate model, while the server leverages the logits offsets between the tuned and untuned surrogates to adjust the outputs of the untuned LLMs and distills the knowledge to update the surrogate model. Experimental results show that FedSFT reduces communication overhead, computation cost, and memory usage by over **4.2×, 9.6×, and 8.3×**, respectively, by fine-tuning OPT-1.3B instead of OPT-13B.

Our contributions are as follows:
1) We propose FedSFT, a novel framework that enables resource-constrained clients to collaboratively fine-tune an LLM without accessing the model parameters or sharing private data.

2) We propose a composite model grounded in an RL-based fine-tuning objective, theoretically showing that the fine-tuned next-token distribution can be expressed as a logit combination of the untuned LLM, the untuned small LM, and the fine-tuned small LM.

3) We develop a novel composite model-driven distillation strategy that effectively guides the small client models in each federated training round.

4) We conduct extensive experiments across diverse models, tasks, and datasets, demonstrating that FedSFT achieves comparable model performance to direct federated fine-tuning of LLMs, while significantly reducing computation, communication, and memory overheads.

**Revisions made in response to reviewer comments.**

For clarity, we use **R** for Reviewers, **W** for Weaknesses, and **Q** for Questions.

(1) Concern on the black-box and full-logits assumptions (R1: W1, Q1; R2: W1).

We clarify that the term ``black-box'' follows its standard usage in machine learning: a model is considered black-box when its internal information—such as parameters, gradients, or architecture—is inaccessible, while its outputs remain accessible. In addition, we conducted new experiments using only the top-5 logits to demonstrate that FedSFT remains effective even when partial logits are available, as is typical for state-of-the-art closed-source models such as GPT-4. As shown in **Table 1**, FedSFT maintains strong performance under this constraint, demonstrating its applicability to real-world closed-source LLMs.

**Table 1: Rouge-L scores (5 seeds) with top-5 logits from the large model for OPT models.
A larger value indicates better performance.**

|Method|Dolly|SelfInst|S-NI|
|--|---|----|---|
|Base (13B)|9.8±0.2|6.7±0.2|7.6±0.1|
|FedIT (13B)|23.6±0.2|15.2±0.7|25.9±0.5|
|FedIT (1.3B)|20.4±0.5|11.5±0.5|21.0±0.1|
|FedIT + SFT (top-5) (1.3B–13B)|23.4±0.3|14.7±0.4|30.7±0.2|
|FedOT (2-2-2)|6.1±0.1|3.7±0.1|3.8±0.1|
|FedSFT (top-5) (1.3B–13B)|**23.7 ± 0.3**|**15.3 ± 0.3**|**31.6 ± 0.3**|

In the revised manuscript, we have added these new experimental results in Appendix C.7.

(2) Concern on baseline fairness and model capacity (R1: W2, Q2).

We clarified that all performance comparisons are made on the same target large LM after tuning, ensuring a fair and capacity-matched evaluation.

(3) Concern on centralized and local baselines (R1: W3, Q4).

We clarified that centralized training violates FL's data-privacy assumptions; nevertheless, we added a centralized upper-bound baseline (CentralSFT) and a local-only baseline (LocalSFT). As shown in Table 2, FedSFT consistently outperforms LocalSFT and closely approaches CentralSFT across all datasets.

**Table 2: Rouge-L scores (5 seeds) for OPT models. A larger value indicates better performance.**
 |Method| Dolly| SelfInst| S-NI|
 |--|-|--|--|
 |LocalSFT (1.3B–13B)|20.7±1.1|11.9 ± 0.8| 20.9 ± 2.1|
 |CentralSFT (1.3B–13B)| 23.7± 0.3| 15.5 ± 0.8| 26.1± 0.2|
 |**FedSFT (1.3B–13B)**|**21.8±0.2**|**15.1±0.6**|**25.8±0.2**|

We have added the experimental results of both the centralized and local baselines in Appendix C.8.

(4) Concern on server-side knowledge distillation is "like giving the model for free to the client,'' (R2: W1).

This is a misunderstanding of our workflow. In FedSFT, distillation is performed entirely on the server, and clients receive only updated small model parameters, which they already possess. Clients never observe the large model’s parameters, logits, hidden states, or any intermediate computations. Since no information from the large model is transmitted to clients, FedSFT does not leak the large model or provide any signal that could be used to reconstruct it.

---

### Note · Authors · 2026-01-03

I have read and agree with the venue's withdrawal policy on behalf of myself and my co-authors.